# Sequencing DNA methylation and hydroxymethylation at co-occurring chromatin features

Rafael de Cesaris Araujo Tavares [1], Somdutta Dhir[1], Xuan He [2], Jack Monahan [3], Minna Taipale[3], Paula Golder[3], Aldo Ciau-Uitz [3], Walraj Gosal[3], David Tannahill [1] & Shankar Balasubramanian [1,2,4]✉

Epigenetic modifications govern chromatin dynamics and cell state. However, current methods cannot simultaneously resolve the presence of multiple DNA modifications at co-occurring chromatin-associated features. It is thus not clear how these features are physically coupled and how their combinations regulate genome function. To address this key question, we report 6-base-CUT&Tag, a method for simultaneous 6-base DNA sequencing at target chromatin features. Using 6-base-CUT&Tag to profile 5-methylcytosine (5mC) and 5-hydroxymethylcytosine (5hmC) at co-occurring histone modifications in mouse embryonic stem cells (mESCs), we identify feature-dependent 5mC/5hmC signatures previously unresolvable with untargeted or bisulfite-based workflows. We show that DNA methylation and hydroxymethylation are specifically coupled with the H3K4me1 mark in mESC enhancers and that H3K4me1-derived signatures robustly distinguish different enhancer functional states.

Chemical modifications of DNA bases and histone tails regulate physical and functional properties of chromatin, including DNA accessibility, genomic architecture, and binding of transcription regulators[1–4]. Methylation of cytosine bases on DNA is critical to establish the epigenetic landscape and governs gene activity and inheritance[5]. Cytosine bases can be methylated by DNA methyltransferases (DNMTs), that transfer a methyl group from the donor S-adenosylmethionine to form 5-methylcytosine (5mC)[6], which can be further oxidised to form 5-hydroxymethylcytosine (5hmC) through the action of the Ten-Eleven Translocation family of proteins (TETs 1-3)[7]. TET enzymes can further oxidise 5hmC to other intermediates like 5-formylcytosine (5fC) and 5-carboxylcytosine (5caC)[8], each of which can be excised by thymine DNA glycosylase, followed by restoration of C, completing a demethylation cycle (Fig. 1a). 5mC is the most abundant cytosine modification (2–4% of all Cs in mouse embryonic stem cells [mESCs]), and 5hmC is the next most abundant cytosine modification (0.1–0.2% of all Cs in mESCs), with 5fC and 5caC being present at substantially lower levels (~0.002% and ~0.0003%, respectively)[8]. While 5hmC can be an intermediate in active DNA demethylation, it has also been shown to be a stable epigenetic mark that is associated with transcriptional activity and cell fate[9–14]. The functional roles of 5mC and 5hmC depend on their genomic context. Broadly, 5mC enrichment at promoters and enhancers is associated with transcriptional repression[15,16], whereas 5hmC is enriched at active regulatory regions—particularly enhancers—and also within the gene bodies of transcriptionally active genes[17,18].

The development of sequencing methods to detect 5mC and 5hmC has been an important focus in epigenomics. While early methods such as bisulfite sequencing[19] cannot distinguish 5hmC from 5mC, later approaches—including oxidative bisulfite sequencing[20], TET-assisted bisulfite sequencing[21], APOBEC-coupled epigenetic sequencing[22], pyridine borane sequencing methods[23,24] and third-generation sequencing platforms (e.g., PacBio, Oxford Nanopore)[25] – have enabled base-resolution mapping of both 5mC and 5hmC. More recent methods have now leveraged state-of-the-art enzymology and

[1]Cancer Research UK Cambridge Institute, University of Cambridge, Li Ka Shing Centre, Robinson Way, Cambridge, UK. [2]Yusuf Hamied Department of Chemistry, University of Cambridge, Cambridge, UK. [3]biomodal, The Trinity Building, Chesterford Research Park, Cambridge, UK. [4]School of Clinical Medicine, University of Cambridge, Cambridge, UK. ✉e-mail: sb10031@cam.ac.uk

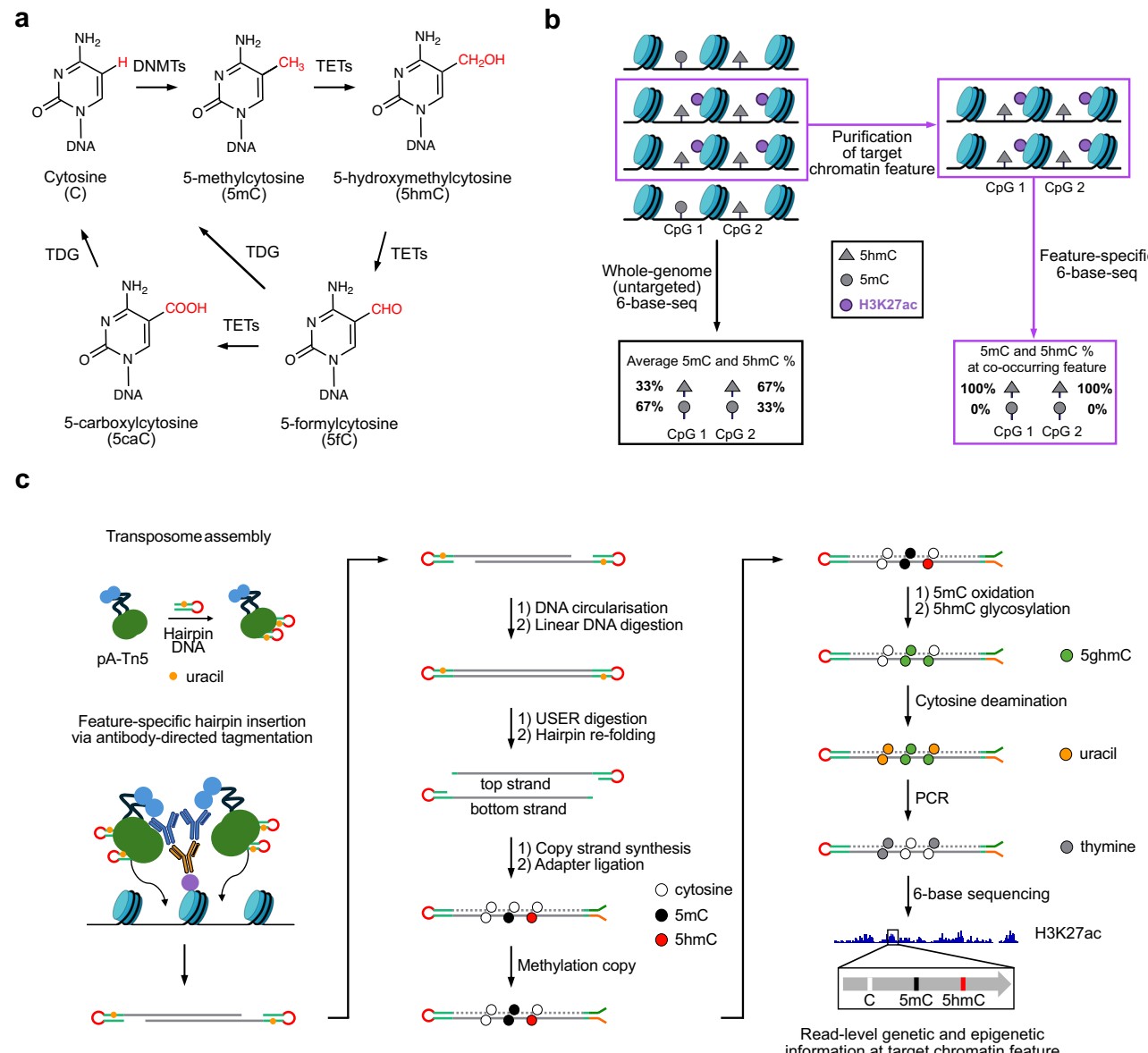

**Fig. 1 | 6-base-CUT&Tag conceptualisation. a** Scheme depicting cytosine (C) methylation by DNA methyltransferase (DNMT) to form 5mC and progressive oxidation by (Ten-Eleven Translocation) TET enzymes to generate 5hmC, 5fC and 5caC modifications. Both 5fC and 5caC can be excised by thymine DNA glycosylase (TDG) to restore C and complete the demethylation cycle. Chemical groups specific to each cytosine state are flagged in red. **b** Conceptual overview illustrating how 'standard' whole-genome 6-base sequencing reports an average ensemble picture that does not accurately represent co-occurring epigenetic features. In contrast, enriching for a specific chromatin feature (e.g., H3K27ac mark) isolates a sub-family of co-occurring protein-DNA fragments and simultaneously reports on 5mC and 5hmC at read level. For simplicity, methylation heterogeneity for each modification is not represented in this scheme. Schematic 5mC and 5hmC percentages are shown for the CpG sites in each case. **c** Complete 6-base-CUT&Tag (6B-C&T) workflow. Specific steps are detailed in the main text (also see Supplementary Fig. 1). An example profile for H3K27ac, 5mC and 5hmC is shown at the bottom of the diagram and exemplifies the coupling of three layers of information originating from the same DNA fragment. Colour code for selected nucleobases is indicated throughout the diagram.

chemistry to enable bisulfite-free and simultaneous detection of 5mC and 5hmC on the same DNA fragment[26–28].

Despite such advancements, a key challenge in the field has been to determine how 5mC and 5hmC are coupled in space and time to other chromatin features such as histone marks, transcription factors and chromatin remodelers. Establishing this relationship is critical to appreciate how epigenetic features are coupled, when and where they co-occur, and how they determine causal relationships and regulatory roles. Current strategies that map DNA modifications and chromatin features largely rely on comparing profiling data from independent experiments for different features. This approach does not necessarily determine the co-occurrence of

these features at the molecular level (Fig. 1b). Earlier attempts to address this limitation have coupled chromatin immunoprecipitation sequencing (ChIP-seq) and cleavage under targets and tagmentation (CUT&Tag) with bisulfite treatment to detect cytosine methylation on DNA fragments directly associated with chromatin proteins[29–32]. However, these methods have major limitations that include substantial loss of DNA caused by bisulfite-mediated degradation[33], which precludes analysis of smaller sample quantities. Crucially, the inability to discriminate between C, 5mC, and 5hmC bases at read level in the same workflow[34] is a major gap in our understanding of how combinations of epigenetic features influence genome regulation.

Here, we introduce 6-base-CUT&Tag sequencing (abbreviated 6B-C&T), a method to simultaneously map G, A, T, C, 5mC, and 5hmC bases at co-occurring chromatin-associated features. 6B-C&T begins with targeted antibody enrichment of specific chromatin features via CUT&Tag that are purified from a heterogeneous pool, followed by an updated 6-base library preparation workflow, sequencing, and analysis (Fig. 1c). This approach substantially reduces sequencing depth requirements and cost relative to whole-genome sequencing analysis (see "Methods", Supplementary Data 1). We demonstrate that 6B-C&T reveals 5mC/5hmC signatures which are feature-dependent and have not been recoverable in previous untargeted or bisulfite-based workflows.

## Results

### 6-base-CUT&Tag (6B-C&T) development

As proof-of-concept for 6-base-CUT&Tag (6B-C&T), we used E14TG2A mouse embryonic stem cells (mESCs) and initially chose H3K27ac, a highly prevalent histone mark in active chromatin, as the feature to profile (Fig. 1c). Using an antibody selective for H3K27ac, we first performed antibody-directed pA-Tn5 tagmentation with a uracil-containing hairpin mosaic-end adaptor, termed ME2U. Tagmented DNA is then subjected to circularisation to generate dumbbell-like molecules that are resistant to exonuclease treatment. Uracil-specific excision reagent (USER) treatment digests the uracil sites of the adaptor, allowing the strands to separate and the residual hairpin sequences on each strand to refold independently. This produces DNA suitable for our 6-base-seq enzymatic conversion workflow[26] (Fig. 1c, Supplementary Fig. 1). After Illumina paired-end sequencing, the four canonical DNA bases and two cytosine epigenetic states can be resolved on individual reads to provide the G, A, T, C, 5mC, and 5hmC sequence context at H3K27ac sites.

An important element in the development of 6B-C&T was to avoid interference from partially tagmented or randomly cleaved DNA, which would be a major source of non-specific background. To address this, we took advantage of the 'scar' sequences left on both strands of the DNA fragment after USER digestion of the inserted ME2U hairpin. These residual sequences are copied and deaminated prior to PCR amplification (Fig. 2a) and thus serve as an internal benchmark to identify valid read pairs derived from double-sided hairpin constructs. Furthermore, circularisation of such constructs makes them resistant to exonuclease digestion, as opposed to non-specifically cleaved or partially tagmented DNA fragments. Motivated by related strategies to remove similar DNA fragments[35,36], we determined that a mixture of T7 Exonuclease (dsDNA-specific), RecJf, and Exonuclease I (ssDNA-specific) achieved the highest recovery of scarred read pairs relative to a protocol with no enzyme treatment or with other enzyme combinations (Fig. 2a).

In our initial H3K27ac 6B-C&T in mESCs, scar-filtered read pairs mapped to regions that strongly overlapped with H3K27ac sites determined independently using standard CUT&Tag (Fig. 2b, c; Supplementary Figs. 2,3). To verify the accuracy of DNA modification measurement, we included three different synthetic spike-ins as fiducials (Methods) that were also subjected to 6-base-seq base conversions alongside tagmented mESC DNA. For each DNA control (5mC-modified Lambda DNA, 5hmC-modified oligonucleotide and unmodified pUC19 DNA), we achieved high sensitivity (98-99.5%) in identifying the true cytosine modification state (Fig. 2d). These results show that 6B-C&T can faithfully recapitulate the genomic distribution of a target feature and simultaneously detect 5mC and 5hmC bases with high accuracy.

Having validated 6B-C&T on H3K27ac, we set out to profile 5mC and 5hmC in mESCs at each of several histone modifications marking distinct types of regulatory elements[37]. For this, we included H3K27ac (active enhancers and promoters), H3K4me3 (promoters), H3K4me1 (enhancers), and H3K27me3 (Polycomb-repressed, bivalent chromatin), along with a non-specific IgG control. As a reference, we also performed whole-genome 6-base-seq, which provides an untargeted

picture representative of the total pool of DNA fragments. Using this approach, we confirmed that the global relative 5mC/5hmC abundance on genomic DNA was consistent with reported values in mESCs (Supplementary Fig. 4a). Genomic enrichments obtained by 6B-C&T were highly reproducible across experimental replicates for each histone mark (Pearson's r ≥ 0.97) and showed good agreement (Pearson's r ≥ 0.86) when compared to standard CUT&Tag (Supplementary Fig. 5). CpG modification levels (i.e. the fraction of CpGs detected as 5mC or 5hmC) at each histone mark also showed excellent reproducibility (Pearson's r ≥ 0.90, Supplementary Fig. 6). Finally, we assessed how well 6B-C&T correlates with CUT&Tag-BS, a method that profiles total cytosine methylation (modC) on CUT&Tag DNA fragments via bisulfite conversion[32]. Both genomic feature enrichment and CpG modification levels for 6B-C&T agreed well with publicly available CUT&Tag-BS data for H3K4me1 and H3K27me3 (Pearson's r ≥ 0.83, Supplementary Fig. 7).

### Measuring 5mC and 5hmC at distinct histone modifications

Methylation is known to be depleted in histone mark-enriched regulatory regions relative to the other regions in the mESC genome[38]. Consistent with this, we observed that average 5mC and 5hmC levels in 6B-C&T were lower than in the whole-genome (untargeted) 6-base-seq experiment (Supplementary Fig. 4a). This difference is due to the 6B-C&T method specifically capturing DNA fragments at histone mark-enriched sites, which are generally less (hydroxy)methylated compared to the whole genome. Furthermore, when looking at individual histone marks, we found that those associated with active chromatin (H3K4me3, H3K27ac, H3K4me1) showed lower overall methylation and a higher 5hmC/5mC ratio than the bivalent/repressed chromatin (H3K27me3) mark, a pattern that was absent in whole-genome data (Supplementary Fig. 4b, Supplementary Fig. 8). Globally, individual CpG sites captured by 6B-C&T mapped to regions with high histone mark enrichment and showed differences in 5mC and 5hmC levels relative to the untargeted method (Supplementary Fig. 9).

We next investigated how 5mC and 5hmC levels in 6B-C&T differed from untargeted 6-base-seq data for the same CpG sites (Supplementary Fig. 10a). First, we identified CpG sites with statistically significant differences between both methods, i.e., differentially methylated CpGs (DMCs) and differentially hydroxymethylated CpGs (DHMCs). While most DHMCs between both methods were located in introns followed by intergenic and promoter regions, DMCs were primarily found in promoters followed by introns/exons (Supplementary Fig. 10b−e). Overall, DMCs predominantly showed lower 5mC levels in 6B-C&T relative to whole-genome (untargeted) data (Supplementary Fig. 10a, Supplementary Fig. 11). This suggests that DNA associated with histone modifications at these sites is mostly hypomethylated relative to the total pool of DNA fragments. This was particularly striking in DMCs co-occurring with H3K4me3 at imprinted gene loci[39] such as *Igfr2*, *Kcnq1ot1* and *Peg3*, suggesting that this active histone mark largely associates with the unmethylated allele (Supplementary Fig. 10a). For DHMCs, active chromatin marks (H3K4me3, H3K27ac and H3K4me1) showed similar numbers of sites with higher or lower 5hmC levels in 6B-C&T relative to the untargeted experiment; in contrast, for the repressive mark (H3K27me3) the majority of DHMCs showed higher 5hmC levels in 6B-C&T than in untargeted 6-base-seq (Supplementary Fig. 10a, Supplementary Fig. 11). Overall, these observations reveal important epigenetic differences between the DNA associated with target histone modifications and the bulk average at the same genomic sites. These quantitative relationships would be difficult to discern from comparative analysis of independent measurements of 5mC, 5hmC, and histone marks.

### Histone mark-specific 5mC and 5hmC enrichment at enhancers

To demonstrate the applicability of 6B-C&T to determine the true base modification status at specific chromatin elements, we analysed a set

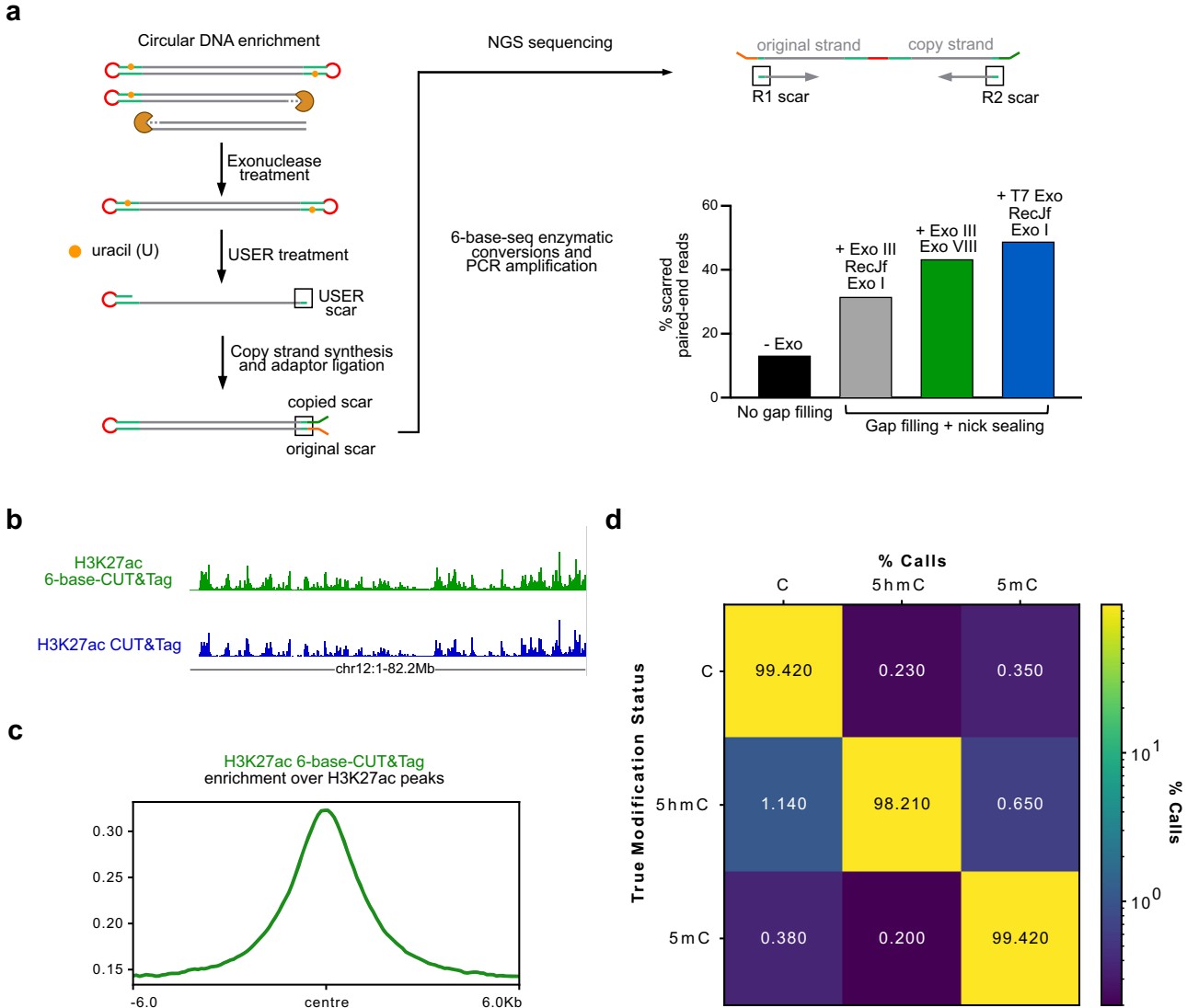

**Fig. 2 | 6-base-CUT&Tag implementation and benchmarking. a** Left: strategy to enrich bona fide 6B-C&T fragments. In this workflow, exonuclease enzymes are first used to digest randomly cleaved or partially tagmented DNA to enrich circularised DNA containing hairpins on both ends. USER digestion is then used to open circular fragments and separate each strand, allowing for hairpin re-folding followed by subsequent copy strand synthesis and ligation of fork-head adaptors. Ligated DNA is now ready for 6-base-seq enzymatic conversions and PCR amplification. After paired-end sequencing, the presence of USER-derived scar sequences on both R1 and R2 reads identifies valid 6-base-seq fragments. Right: percentage of USER-scarred paired-end reads for different exonuclease cocktails tested after gap filling and nick sealing. The protocol without these steps (black) is shown as a control (Methods). Data from H3K27ac profiling in E14TG2A mESCs. **b** Representative genome browser view of 6-base-H3K27ac-CUT&Tag (green) and standard H3K27ac-CUT&Tag (blue) profiles (coordinates shown under the tracks). **c** Metagene plot showing H3K27ac enrichment for 6B-C&T data in peak regions defined using standard H3K27ac-CUT&Tag data. CPM-normalised CUT&Tag coverage is plotted relative to the centre of each consensus peak. **d** Call-rate matrix for 5mC and 5hmC detection in three fiducial DNA controls used in 6B-C&T. Rows represent the true modification state of individual cytosines in each control (C – pUC19 unmethylated DNA; 5hmC – synthetic modified oligonucleotide; 5mC−fully methylated lambda DNA), while columns indicate the experimentally called modification state (calls). Numbers represent the % calls for each column category in each row type. Source data are provided as a Source Data file.

of mESC enhancers that have been independently annotated by the presence or absence of specific histone marks by ChIP-seq[40], including active (H3K27ac, H3K4me1), primed (H3K4me1), and poised (H3K4me1, H3K27me3) enhancers. We observed good agreement at these loci between 6B-C&T and the ChIP-seq data originally used for their annotation (Supplementary Fig. 12). Excellent agreement was also obtained for total DNA methylation levels (%modC) at all enhancer types between 6B-C&T and available C&T-BS data for the general enhancer mark, H3K4me1 (Supplementary Fig. 7c). We next found that different enhancer loci displayed distinct patterns of 5mC and 5hmC enrichment (Fig. 3a, Supplementary Fig. 13). Across all enhancer types, primed enhancers, defined by H3K4me1 alone, generally showed the highest levels of both 5mC (average ~13%) and 5hmC (average ~4%; Fig. 3b). It is notable that this increased enrichment at the H3K4me1 mark was specific to primed enhancers, since other H3K4me1-associated CpG sites showed much lower 5mC and 5hmC levels (Supplementary Fig. 14; Supplementary Data 1−Table 1). In contrast, active and poised enhancers were relatively depleted of both 5mC (~4% for active and ~2% for poised, on average) and 5hmC (~2% for active and poised, on average). For all enhancer types, H3K4me1 consistently showed the highest 5mC and 5hmC levels and the highest 5mC/5hmC ratio among all histone marks (Fig. 3b).

We then assessed whether the enrichment of 5mC and 5hmC at primed enhancers was specific to H3K4me1 by measuring these DNA

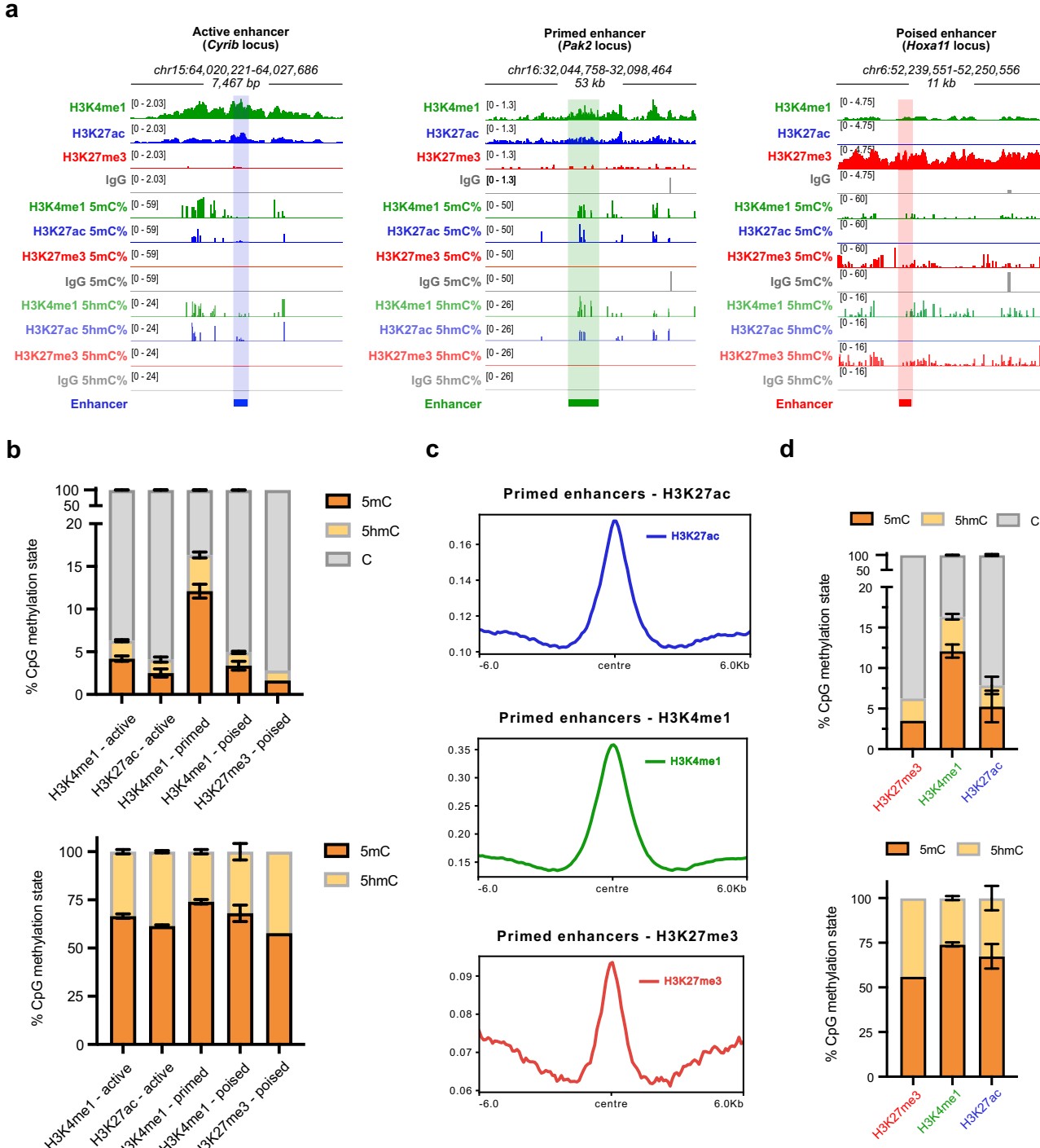

**Fig. 3 | Histone mark-specific 5mC and 5hmC signatures define different mESC enhancer states. a** Genome browser views of representative active, primed and poised enhancer loci. Tracks show histone mark levels and percentages of 5mC and 5hmC obtained with 6B-C&T for enhancer-associated histone marks, in addition to an IgG control. The genomic location of each annotated enhancer is indicated with a horizontal bar and shaded area coloured according to enhancer type (active–blue, primed–green and poised–red). Tracks within each group (CUT&Tag fragment enrichments, %5mC and %5hmC) have the same y-axis scale. **b** Stacked bar plots showing average (mean) relative percentages of CpG methylation states for each enhancer type at different associated histone marks (data from biologically independent replicates: H3K27me3, $N = 2$; H3K27ac, $N = 3$; H3K4me1, $N = 3$). Error bars show standard deviation. The top bar graph shows the total distribution of detected unmodified C, 5mC, and 5hmC, while the bottom graph shows the relative abundances of 5mC and 5hmC. **c** Normalised 6B-C&T read coverages in primed enhancer regions for H3K27ac, H3K4me1 and H3K27me3. Data from one representative experiment plotted relative to the centre of each enhancer region. Replicate information as in (**b**). **d** Stacked bar plots of mean CpG methylation percentages for each histone modification at primed enhancers. Top: all 6-base-seq CpG states (C, 5mC, and 5hmC). Bottom: quantification of the relative abundances of 5mC and 5hmC at each histone mark across biologically independent replicates (H3K27me3, $N = 2$; H3K4me1, $N = 3$; H3K27ac, $N = 3$). Error bars represent standard deviation. Source data are provided as a Source Data file.

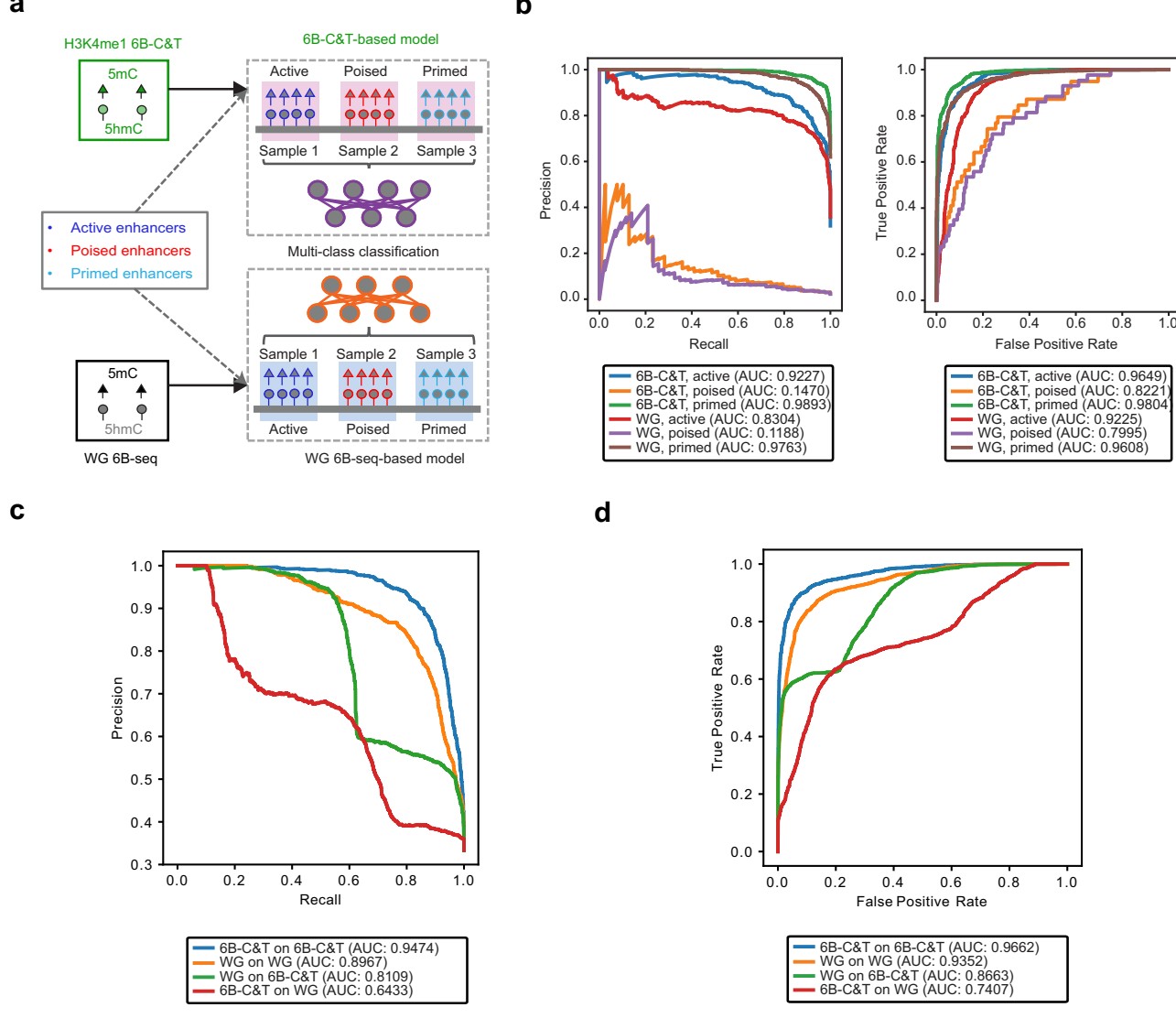

**Fig. 4 | H3K4me1-specific 5mC and 5hmC levels are robust predictors of mESC functional enhancer states. a** Machine learning workflow for enhancer type classification using 6B-C&T (H3K4me1) and whole genome (WG) 6B-seq data. Read-level 5mC and 5hmC counts are extracted from each dataset (see Methods) and used together with reference enhancer annotations (labels) to train a classification model to resolve different enhancer types (samples). Enhancer annotations were obtained from Cruz-Molina et al.[40] **b** Performance evaluation for H3K4me1 6B-C&T-based vs. whole genome 6B-seq models. Precision-recall and Receiver Operating Characteristic curves are shown for each model using chromosome 1 data as a test set (not included in training). **c** Multi-class precision-recall curves for different training vs. test dataset combinations (e.g., WG on 6B-C&T: trained on whole genome data and tested on 6B-C&T data). All tests were performed on chromosome 1 data (not included in training) using the trained model for classification into different enhancer types. **d** Multi-class Receiver Operating Characteristic curves for the machine learning routine described in (**c**). Colour legend and area under the curve (AUC) values are shown below each graph. Source data are provided as a Source Data file.

modifications at other enhancer-associated histone marks potentially present at these sites, i.e., H3K27ac and H3K27me3. We first confirmed that annotated primed enhancers were enriched with the H3K4me1 mark but also showed appreciable levels of H3K27ac and H3K27me3 (Fig. 3c, Supplementary Fig. 15). Across all three histone marks, H3K4me1 displayed the highest fraction of both 5mC and 5hmC, and the highest 5mC/5hmC ratio (Fig. 3d, Supplementary Fig. 16). This pattern, which cannot be detected with previously reported methods like ChIP-BS[41], suggests that H3K4me1-marked nucleosomes are sites for preferential 5mC retention at enhancers. Active (H3K27ac) and bivalent (H3K27me3) histone marks, which do not represent the primed enhancer state, showed a greater tendency towards demethylation (lower overall methylation levels, higher 5hmC/5mC ratio). These results suggest that 5mC and 5hmC are coupled to different extents with each chromatin mark and, more generally, that different

enhancer states are defined by distinct combinations of epigenetic modifications.

## 5mC and 5hmC at H3K4me1 resolve distinct enhancer functional states

H3K4me1 is common to all enhancer functional states, so we next asked whether 6B-C&T methylation at this single feature better resolves different enhancer types compared to the more heterogenous whole genome (untargeted) 6-base-seq data. For this, we used H3K4me1 6B-C&T data to train a machine learning model for classification of enhancer types (active, primed and poised) and compared its performance to a model with the same architecture trained on whole-genome 6B-seq data (Fig. 4a, Supplementary Fig. 17). For all enhancer types, the 6B-C&T-derived model showed superior performance when compared to the whole genome-derived model (Fig. 4b–d;

Supplementary Fig. 18). This demonstrates that the H3K4me1-coupled 5mC and 5hmC signal more robustly resolves bona fide enhancer states than its unenriched equivalent. These results show that 6B-C&T can be used to evaluate co-occurring epigenetic features and retrieve improved information on individual molecular states (Fig. 1b).

## Discussion

6B-C&T detects 5mC and 5hmC bases that co-occur with target chromatin features. Profiling of major regulatory histone modifications, such as H3K4me1 and H3K27me3, reveals patterns that are consistent with other measurements of total DNA methylation at these marks[32,41,42] and adds 5hmC as a new dimension to the multiomics analysis of individual DNA fragments. Interrogation of feature-specific 5mC/5hmC signatures provides a higher-resolution framework for mechanistic dissection of methylation dynamics at transcriptional regulatory elements.

To exemplify the utility of 6B-C&T we used it to dissect the epigenetic heterogeneity of mESC enhancers and found that the accumulation of 5mC and 5hmC in these regions depends on the associated histone mark. Furthermore, although all enhancer functional states are marked by H3K4me1, the co-occurrence of 5mC and 5hmC on the same DNA fragment with this histone modification had never been measured. Here, we show that both DNA marks co-occur at significantly higher levels with H3K4me1 than with other histone marks. Our machine learning analysis shows how these different layers of epigenetic information (i.e., histone and DNA modifications) are coupled and intrinsically linked with enhancer functional states. Importantly, the use of H3K4me1-specific methylation signatures enables the identification of functional enhancer states without the need for whole-genome sequencing or multiple histone mark enrichment profiles.

Our work allows for the precise measurement of coupled epigenetic features at individual loci and has the potential to largely expand the combinatorial epigenetic code. Discriminating at read level between multiple combinations of epigenetic marks will be paramount to dissect their inherently different functional roles and mechanisms. Our method also has the practical advantage of lower sequencing depth and cost, as compared to multiple whole-genome sequencing runs. We envisage routine application of 6B-C&T will provide new insights into how different epigenetic layers combine to modulate chromatin behaviour and cellular phenotype.

## Methods

### Cell culture

E14TG2A mouse embryonic stem cells were cultured in serum/LIF conditions on gelatin-coated plates. Serum/LIF medium was prepared with high-glucose DMEM (Sigma, D6546-500mL) supplemented with 10% FBS (Gibco, 16141079), 1X GlutaMax-I (Gibco, 35050-038), 1X NEAA (Gibco, 11140-035), 0.1 mM β-mercaptoethanol (Sigma, M3148) and 1000 U/mL LIF (PeproTech, murine LIF, 250-02-25UG). For maintenance, cells were grown until 70–80% confluence and subcultured by dissociation with StemPro Accutase Cell Dissociation Reagent (Invitrogen A1110501). Cultures were routinely monitored and tested negative for *Mycoplasma* contamination. E14TG2A cells were a gift from Wolf Reik (Babraham Institute, Cambridge).

### Preparation of tagmentation adaptors

Tn5ME-A, Tn5ME-B and Tn5MErev oligonucleotides were reconstituted to 200 μM in annealing buffer (10 mM Tris-HCl pH 8.0, 50 mM NaCl, 1 mM EDTA). Tn5ME-A and Tn5ME-B were mixed separately with equal amounts of Tn5MErev and annealed on a thermocycler (95 °C for 2 min, then cooled in 5 °C steps with 5 min incubation at each step until 25 °C). The resulting mosaic-end adaptors are termed MEDS-A and MEDS-B, respectively. The ME2U hairpin adaptor was reconstituted to 25 μM in Nuclease-Free Duplex Buffer (30 mM HEPES, pH 7.5; 100 mM

potassium acetate, IDT11-01-03-01) and annealed by heat denaturation at 95 °C for 2 min, followed by slow cooling at 0.1 °C/second to 10 °C. All oligonucleotide sequences (including chemical modifications) are listed in Supplementary Data 1 (Table 2).

### pA-Tn5 transposome assembly

pA-Tn5 transposase was prepared in-house[43] or obtained commercially (Active Motif, 53162) and loaded with application-specific DNA adaptors. For the standard CUT&Tag workflow, 50 μL of pA-Tn5 (4 μM) were mixed with a threefold excess of each pre-annealed mosaic-end adaptor, MEDS-A and MEDS-B, and rotated at room temperature for 1 h. For 6-base-CUT&Tag, 50 μL of pA-Tn5 (4 μM) were mixed with a threefold excess of folded ME2U hairpin adaptor and similarly rotated at room temperature for 1 h. Transposome samples were stored at −20 °C.

### Standard CUT&Tag (cleavage under targets and tagmentation)

Standard CUT&Tag was performed as in Kaya-Okur et al.[44] with modifications. Briefly, *Dynabeads MyOne Streptavidin T1* beads (Invitrogen 65601) were conjugated with biotinylated ConA[45] (Sigma C2272) and activated in binding buffer (20 mM HEPES pH 7.5, 10 mM KCl, 1 mM CaCl$_2$, 1 mM MnCl$_2$) prior to cell binding. 500,000 cells per assay were lightly fixed for 2 min with 0.1% formaldehyde in PBS and fixation was stopped by adding glycine to a final concentration of 120 mM. Fixed cells were resuspended in wash buffer (20 mM HEPES pH 7.5, 150 mM KCl, 0.5 mM spermidine, and Complete EDTA-Free Protease Inhibitor, Roche 11873580001) and immediately bound to ConA-conjugated *MyOne T1* beads. Primary antibody binding was performed overnight at 4 °C in wash buffer supplemented with 0.05% digitonin (Sigma 300410-250MG), 2 mM EDTA, 0.1% BSA and 1:50 dilutions of antibody stocks (H3K27ac—Abcam, ab4729-100ug; H3K4me1—Abcam, ab8895; H3K4me3—Abcam, ab8580; H3K27me3—Cell Signalling Technology, 9733; normal rabbit IgG—Cell Signalling Technology, 2729). For secondary antibody binding, a guinea pig anti-rabbit IgG (Antibodies-Online ABIN101961:2 μg) was diluted 1:25 in dig-wash buffer (wash buffer + 0.05% digitonin) and binding performed at 25 °C for one hour. pA-Tn5-MEDS-A/B transposome was diluted 1:250 in dig-300 buffer (20 mM HEPES pH 7.5, 300 mM KCl, 0.5 mM spermidine, 0.01 % digitonin, and Complete EDTA-Free Protease Inhibitor) and allowed to bind at 25 °C for one hour. Feature-specific tagmentation was done in dig-300 buffer supplemented with 10 mM MgCl$_2$ at 37 °C for one hour and stopped by washing cells with TAPS wash buffer (10 mM TAPS pH 8.5, ThermoScientific J63268.AE + 0.2 mM EDTA). DNA was released and cross links were reversed by Proteinase K treatment at 55 °C for one hour (0.5 mg/mL Proteinase K Invitrogen EO0491, 0.5% SDS, 10 mM Tris-HCl pH 8.0), followed by clean-up with Zymo DNA Clean and Concentrator (Zymo Research D4013). PCR was carried out using Q5 Hot Start High-Fidelity 2X Master Mix (M0494S) and indexed primers (Supplementary Data 1 −Table 2) on a thermocycler (programme: 72 °C for 5 min, 98 °C 30 s; 12 cycles of 98 °C 10 s, 63 °C 10 s; final extension at 72 °C 1 min; 4 °C hold). PCR products were purified by double-sided size selection with AMPure XP beads (Beckman Coulter A63880, 0.4X followed by 1.3X). QC was performed on a 4200 TapeStation (Agilent) and libraries were quantified using the NEBNext library quantification kit (E7630L). Libraries were sequenced on Illumina's NextSeq 2000 sequencing system (paired-end 2 ×151-bp, dual index 8-bp).

### 6-base-CUT&Tag (6B-C&T)

**Preparation of tagmented DNA for 6-base sequencing.** Feature-specific tagmentation was performed as in standard CUT&Tag, with modifications. Since the size selection and 6-base enzymatic conversion steps reduce DNA recovery, we increased input material to maintain DNA yields and PCR duplication rates comparable to conventional CUT&Tag. Four CUT&Tag samples (500,000 cells each) were

prepared in parallel per biological sample. pA-Tn5-ME2U transposome was diluted 1:25 in dig-300 buffer and bound to cells for one hour at 25 °C, followed by tagmentation. Proteinase K digestion was performed at 37 °C for one hour and DNA was purified on Zymo DNA Clean and Concentrator. Eluted DNA from all four samples (i.e., 4 ×500,000 cells) was pooled and subjected to two rounds of right-side size selection (0.4X) with SPRI magnetic beads (*duet multiomics solution evoC* kit, biomodal) and Zymo column purification to remove untagged genomic DNA. 4 pg of long DNA controls (tagmented unmethylated pUC19 and methylated Lambda DNA) were added to each sample at this stage. Gap repair was carried out with 8 U Phusion DNA polymerase (M0530S) and 320 U *Taq* DNA ligase (M0208S), in 1X *Taq* DNA ligase reaction buffer containing 0.8 mM dNTPs (NEB N0447S) at 37 °C for one hour. DNA was purified on Zymo columns and digested in exonuclease digestion mix (1X NEB buffer 4 NEB B7004S, 40 U T7 Exonuclease NEB M0263S, 120 U RecJf NEB M0264S, 80 U Exonuclease I from *E. coli* NEB M0293S) for 4 h at 37 °C, followed by an additional round of Zymo column clean-up.

For optimisation of exonuclease digestion conditions, a range of enzymes were tested to specifically degrade linear dsDNA, in addition to the ones employed in the final workflow (see above). For tests with Exonuclease III (NEB M0206S), 1X rCutSMART (NEB B6004S) was used as the reaction buffer along with 100 U of Exo III per 500,000-cell equivalent of CUT&Tag material. For tests with Exonuclease VIII-truncated (NEB M0545S), 10 U were used per 500,000-cell equivalent of CUT&Tag material, also in rCutSMART buffer.

**6-base-seq library preparation.** The following steps were performed with early access reagents from the *duet multiomics solution evoC* kit (biomodal). To digest ME2U hairpin termini, 23.7 μL of exonuclease-treated DNA (enriched for circular dumbbell-like fragments) were treated with 3 μL of HD enzyme (hairpin digestion module) and 3 μL HD buffer for 30 min at 37 °C, followed by 1.8X SPRI clean-up (elution = 14 μL water). DNA was denatured at 95 °C for 2 min and slowly cooled (−0.1 °C/s) to 10 °C to enable strand separation and hairpin re-folding. At this stage, 0.4 pg short hmC control oligonucleotide (spike-in) was added to the sample. Strand copy synthesis (SC module) was set up with 13 μL DNA, 2.5 μL SC buffer, 2 μL dNTP mix, 2 μL SC enzyme 1, 2 μL SC enzyme 2, 2.5 μL hmCP additive and 1 μL hmCP enzyme, incubated at 37 °C for 30 min, followed by denaturation at 95 °C for 2 min and slow cooling (−0.1 °C/s) to 10 °C. 2.5 μL fork-head adaptor were then added to the DNA sample, along with 0.5 μL hairpin ligation additive and 15 μL hairpin ligation mastermix, and the sample was incubated at 20 °C for 15 min and cleaned up with 1.2X SPRI beads (elution = 15 μL water). After ligation, the methyltransferase (MT) module was used to methylate the copy strand at unprotected 5mCpG sites. DNA from the previous step was mixed with 11.6 μL MT buffer, 0.4 μL MT additive 1, 1 μL MT additive 2 and 13 μL freshly reconstituted MT enzyme, incubated at 23 °C for one hour, then at 65 °C for 15 min, and cleaned up with 1.2X SPRI beads (elution = 31 μL water). 30 μL MT-treated DNA were treated with oxidation (Ox) mix (10 μL Ox additive 1 solution, made by reconstituting solid Ox additive 1 in 400 μL Ox buffer; 1 μL DTT solution; 1 μL hmCP additive; 1 μL hmCP enzyme and 2 μL Ox enzyme), followed by addition of 1 μL of 1:1250 dilution of Ox additive 2. Samples were incubated at 37 °C for one hour and cleaned up with 1.8X SPRI beads (elution = 31 μL water). Deamination (DA module) was performed on 30 μL DNA by adding 12.7 μL water, 17.5 μL DA buffer, 1.8 μL ATP, 3.5 μL MgCl$_2$, 2 μL DA enzyme 1 and 2.5 μL DA enzyme 2 and incubating at 37 °C for 90 min. Samples were purified by adding 6.5 μL DA clean-up reagent and concentration with 1.2X SPRI beads (elution = 21 μL water). 20 μL DNA were then PCR-amplified with 25 μL PCR mastermix and 5 μL UDI primers (programme: 98 °C 30 s; 12 cycles of 98 °C 10 s, 62 °C 30 s, 65 °C 60 s; final extension at 65 °C 5 min; 4 °C hold). DNA was purified with 0.9X SPRI beads (elution = 14 μL *evoC* dilution buffer), analysed on a 4200 TapeStation and quantified with the NEBNext library quantification kit. Sequencing was performed on the NextSeq 2000 platform (paired-end 2 ×151-bp, dual index 8-bp).

**6B-C&T workflow without gap repair and exonuclease digestion**
We initially developed 6B-C&T using a strategy for end repair of hairpin-containing tagmented DNA which slightly differs from the final optimized workflow (Supplementary Fig. 1). After two rounds of SPRI removal of untagged material, DNA was end-repaired via strand displacement of the inserted hairpin and gaps were filled in using Phi29 DNA polymerase (NEB M0269S). The reaction (25 μL) was set up with SPRI-eluted DNA, 1X Phi29 DNA polymerase reaction buffer, 0.2 mM dNTPs, 0.3 U/μL Phi29 DNA Pol and 0.1 mg/mL albumin, and incubated at 30 °C for 30 min, then 65 °C for 10 min, followed by Zymo column clean-up. Eluted DNA was then subjected to USER treatment (HD enzyme from the *duet multiomics solution evoC* kit) to digest the original ME2U hairpin strands, followed by standard 6-base library construction.

**Generation of ground-truth DNA methylation controls**
For the 5hmC control, we used 80 bp oligonucleotides (ATDbio, UK)[26] that were either hydroxymethylated or unmodified at specific CpGs (Supplementary Data 1−Table 2). Here, short synthetic oligonucleotides were first annealed in appropriate pairs and quantified using a dsDNA Qubit Quantification Assay kit (Qiagen). Then, 100 ng of the annealed oligonucleotide were ligated to synthetic ME2U hairpin adaptors. The construct was treated with USER enzyme and purified using SPRI beads.

For long genomic DNA controls, 40 ng of enzymatically methylated (methylated using a CpG methyltransferase) bacteriophage λ DNA (EpigenDX) were added to an equal proportion of unmethylated pUC19 isolated from a methylation-negative strain of Escherichia coli (Dam−, Dcm−), and quantified using a dsDNA Qubit Quantification Assay kit (Qiagen). This mixture was tagmented using Tn5 preloaded with the ME2U hairpin. The tagmented DNA was purified using SPRI beads.

**Whole-genome 6-base-seq using Tn5 tagmentation**
**Sample preparation and genomic DNA isolation.** E14TG2A cells were harvested, spun down and washed once with PBS + 0.1% BSA. After pelleting, the supernatant was removed and cells were stored at −20 °C until genomic DNA extraction. Genomic DNA isolation was carried out using the Quick-DNA MicroPrep Kit (Zymo Research Cat No D3020) following manufacturer's instructions. Purified DNA was quantified using Qubit and stored at −20 °C.

**Tagmentation.** To perform whole-genome Tn5-based 6-base-seq, the ME1U hairpin adaptor (Supplementary Data 1−Table 2) was annealed in Nuclease-Free Duplex Buffer (30 mM HEPES, pH 7.5; 100 mM potassium acetate, IDT11-01-03-01) at 25 μM and then complexed in a 1:1 ratio with Tn5 in TPS buffer (Creative Enzymes Cat No NATE-1629). 12 pmol of loaded Tn5 were used to tagment 80 ng of genomic DNA. Tagmentation was carried out in LM Buffer (Creative Enzymes Cat No NATE-1629) for 10 min at 56 °C. Before DNA purification, DNA was released by adding SDS to a final concentration of 0.05% and incubating at 55 °C for 15 min. After incubation, DNA was purified with 1.8X SPRI beads and eluted in 16.5 μL of nuclease-free water. 1 μL of eluted DNA was analysed on a TapeStation to determine the tagmentation efficiency.

**6-base-seq library prep and sequencing.** 15.5 μL tagmented DNA were first gap-filled with 8U of Phi29 DNA polymerase (NEB Cat No M0269) in Phi29 reaction buffer (1X Phi29 buffer, 200 μM dNTPs, 100 μg/ml albumin) by incubating at 30 °C for 30 min, followed by denaturation at 65 °C for 10 min. DNA was then purified with 0.9X SPRI beads and subjected to USER (NEB Cat No 5505) digestion (in 1X

rCutSmart buffer, NEB B6004S) to cleave the original ME1U hairpin strand. After USER reaction, DNA was purified with 1.8X SPRI beads, eluted with 12 μL library dilution buffer (biomodal) and then denatured at 95 °C for 2 min followed by slow cooling (0.1 °C/s) to 10 °C. Resulting DNA was processed using the *duet multiomics solution evoC* kit (biomodal) according to the protocol described for 6B-C&T and amplified with six PCR cycles. After PCR, samples were purified with 0.9X SPRI beads, eluted in dilution buffer (biomodal), quantified with Qubit and analysed by Tapestation. Sequencing was performed on a NovaSeq 6000 platform (paired-end 2 ×151-bp, dual index 8-bp).

### Bioinformatic analysis and data visualisation

**Standard CUT&Tag data analysis.** Libraries were demultiplexed using *demuxFQ* (flags: -c -d -i -e -t 1 -r 0.01 -R -l 9). *FastQC-* version 0.11.8 was used to assess the read quality. Bases with a quality score below 20 were trimmed from both reads using *Cutadapt* (cutadapt -q 20). Reads were aligned to the GRCm38/mm10 reference genome using *bwa mem* with default parameters. Duplicates were removed using *Picard* version 2.20.3 (*Picard MarkDuplicates*). BigWig files were created on the deduplicated BAM file, using *deepTools bamCoverage* (parameters: --normalizeUsing CPM, --binSize 10). Peaks were called using *Seacr* version 1.3 without input control reporting the top 1% by AUC regions, using both relaxed and stringent criteria. For consensus peak calling, overlap across biological replicates was calculated with intervene tools generating a series of .bed files. Peaks that were common to at least 2 of the 3 biological replicates were considered as consensus peaks. Metagene profiles were generated using *deepTools*.

**6-base sequencing data pre-processing.** FASTQs from 6-base-seq libraries were processed through a modified version of biomodal's duet analysis pipeline. For 6B-C&T (ME2U hairpin), 9-nt scar sequences, AAGAGATAG (R1) and AAAAAACAA (R2), derived from the deaminated, USER-treated Tn5 mosaic end (ME) sequence, were trimmed from the 5′ ends of reads 1 and 2, respectively. Read pairs that lacked either scar sequence were discarded. Adaptor sequences containing the full 19 nt Tn5 ME sequence and 2 uracils (ME2U) hairpin were trimmed from the 3′ ends of reads 1 and 2. For whole-genome 6-base-seq (ME1U hairpin), NNNNNNNNNAGATGTGTATAAGAGATAG (R1) & NNNNNNNNNAAATATATATAAAAAACAA (R2) were trimmed from the 5′ ends of reads 1 and 2, respectively. Similarly, read pairs that lacked either scar sequence were discarded. Adaptor sequences containing the full 19 nt Tn5 ME sequence and 1 uracil (ME1U) hairpin were trimmed from the 3′ ends of reads 1 and 2.

In all cases, a further 9 bp were trimmed from the 3′ ends of reads 1 and 2 to remove the gaps generated during Tn5 transposition. Trimmed read pairs were then resolved using the bespoke resolution algorithm from Füllgrabe et al[26]. The resolved, single-end reads were aligned against a GRCm38/mm10 reference (gencode vM25) that contained additional sequences for pUC19, bacteriophage lambda and short spike-in oligonucleotides using *bwa-mem2* (v2.2.1). Multi-mapping reads, secondary and supplementary alignments were filtered out. DNA (hydroxy)methylation was quantified per aligned position from MM tags as part of the duet analysis pipeline. Compiled sequencing data analysis reports for all datasets were obtained on NextFlow and are available in Supplementary Data 1 (Tables 3–7).

**Call-rate matrix for 6B-C&T.** A call-rate matrix was used (Fig. 1g) to measure the accuracy of 5mC and 5hmC calls. Each cell in the matrix m represents the rate at which the method calls a particular modification state X when the true modification is Y. For example, the cell m(5mC, C) represents the rate at which the method calls unmodC when the true modification status is 5mC. Each row of the matrix corresponds to the rate at which each modification status is called for a particular true modification status. The matrix was estimated using three different spike-in controls: fully unmethylated pUC19 DNA for the row

corresponding to a true state of unmethylated C; fully methylated lambda for the row corresponding to a true state of 5mC; and synthetic oligonucleotide for the row corresponding to a true state of 5hmC. For a given row, the rate is calculated as the proportion of bases with each modification status in the set of bases for which the genetic base call is C and which are aligned to CpGs in the given control sequence.

**Analysis and visualisation of 6B-C&T and WG 6B-seq data.** Individual 5mC and 5hmC percentages were calculated at each CpG site for all replicates. For downstream analyses, only cytosines with a read coverage ≥ 20 were included, except for genome browser visualisation in IGV, where a minimum coverage of ≥ 10 was used. BigWig tracks for 6B-C&T–derived 5mC and 5hmC across the four histone marks were generated by merging individual replicates. Merged bedGraph files were created using *bedtools* (v2.31.0), and BigWig files for IGV visualisation were produced using *bedGraphToBigWig*.

Average 5mC and 5hmC percentages were calculated for each replicate of the 6B-C&T datasets (across the four histone marks) and the whole-genome 6-base-seq dataset using a custom R script. Differential methylation analyses, including generation of volcano plots, were performed for each 6B-C&T dataset relative to whole-genome 6-base-seq using *methylKit* (v1.34.0). Window-based analyses were conducted with the *methylKit* tiling function (win.size = 100; step.size = 100). Cytosines with read coverage <20 were excluded during *methylKit* processing. Differentially methylated positions were identified using the *calculateDiffMeth* function. Regions with $p < 0.05$, $|log2FC| \geq 0.5$, and an absolute methylation difference ≥15% between groups were defined as differentially methylated regions (DMRs).

Shared CpG enrichment profiles were generated using a custom bash workflow. The GRCm38/mm10 genome was partitioned into 1 kb bins, and average enrichment values for each histone mark were computed across these bins using standard CUT&Tag datasets. For CpGs shared between whole-genome 6-base-seq and 6B-C&T datasets, cytosine modification percentages were plotted as a function of histone mark enrichment ($log_2$ scale) for each histone mark.

Average read profiles for 6B-C&T centred on each enhancer class were generated using *deepTools*. Briefly, *bamCoverage* was used to produce BigWig files from BAM files containing 6B-C&T–resolved reads, using a bin size of 20. Peak density distributions were computed with *computeMatrix* (100 bp bins) and visualised with *plotProfile*. The enhancer analysis for global distributions of 5mC and 5hmC was performed using a custom R script (v4.3.1). Briefly, all CpG sites overlapping with enhancer peaks (primed, poised, or active, defined in Cruz-Molina et al.[40]) were identified with *BEDTools* and levels of each CpG state (C, 5mC and 5hmC) at each enhancer were calculated and visualised using R and GraphPad Prism.

Publicly available CUT&Tag-BS data for mESCs were downloaded as raw FASTQ files from the European Nucleotide Archive (ENA), EMBL. Adaptor sequences were removed using *Cutadapt* with the parameters "-a CTGTCTCTTATACAC -A CTGTCTCTTATACAC -O 5 -q 0 -m 20 -p". The trimmed reads were subsequently aligned to GRCm38/mm10 genome using *Bismark* v0.24.044[46] with parameters "-X 1000 --non_bs_mm". Publicly available ChIP-seq datasets for wild-type mESCs were downloaded from GEO. Raw reads were filtered to remove those with a mean base quality score <20 and aligned to the mouse genome using BWA. The resulting BAM files were processed, and peak calling was performed with *MACS2*[47] with parameters $p \leq 0.01$; Fold-enrichment ≥ 2 and Broad Region Calling ON.

Distributions of 5mC and 5hmC levels across genomic windows (corresponding to IGV snapshots) were visualized using a custom Python workflow that parsed TSV input files and removed invalid entries. Boxplots comparing methylation values across each enhancer mark and IgG were generated with matplotlib, using a fixed colour palette and ordering groups by their median values.

For differential methylation scatter plots, DSS-like estimates of mean CpG methylation for the two treatment groups (6B-C&T and WG 6B-seq), along with dispersion parameters, Wald statistics, and FDR-adjusted q-values, were derived from the *calculateDiffMeth* output of the *methylkit* package. These metrics were used to visualise concordance between treatment methylation means ($\mu_1$ vs $\mu_2$) by generating jittered, rasterized scatter plots using *ggplot2*.

Pearson correlations of modified CpG fractions between 6B-C&T and other libraries were computed for CpGs with ≥20× coverage. Pairwise correlations within samples were obtained using the *getCorrelation* function from the *methylKit* package.

Chemical structures in Fig. 1a were generated using *ChemDraw®*. IGV[48] was used for data visualisation in Fig. 1c, Fig. 2b and Fig. 3a. Plots in Fig. 3b, Fig. 3d, Supplementary Fig. 4 and Supplementary Fig. 16 were generated with Prism (GraphPad Software, Massachusetts USA).

### Machine learning analysis
**Data preparation.** Raw counts were collected for each methylation state (C, 5mC, 5hmC and ambiguous modC, (i.e., modified C sites that cannot be resolved into 5mC or 5hmC) from 6-base-seq data (6B-C&T or WG 6B-seq). Raw counts were then normalised by the total counts at each CpG, and then the average was taken across experimental replicates. CpG sites mapping to enhancer regions were collected, and the sequential CpG methylation states in each enhancer region were organised as a sample.

**Model architecture.** As H3K4me1 is the common feature of annotated active, poised, and primed enhancers, we used the samples curated from methylation profiles obtained from H3K4me1 6B-C&T to train a model for classification into these three enhancer classes. To further address the advantages of using H3K4me1-enriched 5mC and 5hmC information, we also trained an independent model with the same architecture based on the methylation profiles generated by whole genome 6B-seq. The model architecture (Supplementary Fig. 17) consists of a linear projection layer that projects the normalised CpG methylation states to a 32-dimension vector (*d_model*). This projection layer was followed by a self-attention layer, which consisted of four attention heads (*num_head*) and a two-layer feed-forward network with 256 hidden units (*ff_dim*), and was used to extract the features for the final classification layer. The classification layer is a two-layer fully connected neural network with 256 hidden units (*mlp_h_dim*) for each layer. The 'elu' activation function was used for non-linearity. A dropout probability of 0.2 (*dropp*) was set for avoiding overfitting. CpG methylation profiles from chromosome 1 were retained for model evaluation.

**Enhancer classification.** Enhancer regions were defined and classified into three categories (active, poised, and primed) according to a previous study[40]. Here, we designed a machine learning model to predict the type of each enhancer based on its 6-base methylation profile. Specifically, for a set of enhancers $\mathbf{E} = \{\mathbf{e_1}, \mathbf{e_2}, \ldots, \mathbf{e_n}\}$, each enhancer can be represented as a sequential vector of normalised signals {*num_ambiguous_modC, mC, hmC, C*} for its CpG loci. Specifically, $\mathbf{e_i} = [\mathbf{m_1^i}, \mathbf{m_2^i}, \ldots, \mathbf{m_k^i}]$, where $m_j^i$ is the $j$-th CpG locus in the $i$-th enhancer, given by $\mathbf{m_j^i} = [x_{ambiguous\_modC}, x_{mC}, x_{hmC}, x_C]$, and $x$ represents the normalised signal. The enhancer labels can be noted as $\mathbf{Y} \in \{0, 1, 2\}^n$, where 0 means active, 1 means poised, and 2 means primed. The classification task is then to find a classifier $f : \mathbf{e} \rightarrow \mathbf{y}$ to minimize the expected loss:

$$f^* = \mathrm{argmin}_{f \in F} E[L(f(\mathbf{e}), \mathbf{y})]$$

Where $L(\cdot)$ is a loss function, we used the cross-entropy loss for the classification task.

**Self-attention.** To capture the associations among CpG methylation states in enhancers, we applied a self-attention module to the model[49]. After projection to a high-dimension space by the linear projector, the CpG methylation profile is simultaneously used as a query, key, and value matrix to calculate the attention matrix and the weighted value. The weighted value is then converted to informative embedding, which is further used for classification.

### Statistics & reproducibility
Experiments were performed independently with different E14TG2A cultures ($N = 2$ for H3K27me3 6B-C&T; $N = 3$ for H3K27ac 6B-C&T; $N = 3$ for H3K4me1 6B-C&T; $N = 3$ for H3K4me3 6B-C&T; $N = 3$ for IgG 6B-C&T and $N = 2$ for WG 6B-seq), and no statistical method was used to pre-determine sample size. Pearson correlation coefficient, representing effect size, was used to verify reproducibility. No data were excluded from any analyses. Descriptive statistics were computed using GraphPad Prism 10 (GraphPad) or R. Randomization or blinding were not required nor performed in this study. Differences between groups were assessed in Supplementary Fig. 10 using *methylKit*, which employs a logistic regression test for $p$ value calculation, and in Supplementary Fig. 14 using a Wilcoxon rank sum test (results in Supplementary Data 1–Table 1).

### Reporting summary
Further information on research design is available in the Nature Portfolio Reporting Summary linked to this article.

## Data availability
Sequencing data generated in this study is available through NCBI Gene Expression Omnibus under accession code GSE296587, with no access restrictions. Publicly available mouse ESC CUT&Tag-BS data were obtained from GEO under accession code GSE179266 and ChIP-seq datasets were obtained from GEO under accession code GSE89211. Source data are provided with this paper.

## Code availability
Code is available on GitHub at https://github.com/sblab-informatics/6B-CUTnTag, with no access restrictions.

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

## Acknowledgements

We thank staff from the Research Instrumentation and Cell Services and the Genomics core facilities at the Cancer Research UK Cambridge Institute for research support. We thank previous and former members of the Balasubramanian group for helpful discussions on the development and application of 6B-C&T, especially Alice Dubois-Veltman, Zutao Yu, Sean M Flynn, and Larry Melidis. The Balasubramanian laboratory is supported by CRUK core funding (SEBINT-2024/100003 to S.B.), CRUK programme funding (C9681/A29214 to S.B.), and University of Cambridge Herchel Smith funds (S.B.). S.B. is a Wellcome Trust Investigator (209441/Z/17/Z).

## Author contributions

R.C.A.T., D.T., and S.B. conceived the study. R.C.A.T. led the study, developed and optimised the 6B-C&T workflow, with wet-lab and computational input from S.D., J.M., M.T., P.G., A.C-U., and W.G. S.D. performed bioinformatic analyses with input from J.M. X.H. performed the machine learning analysis. All authors contributed to interpreting the

results and provided critical feedback. R.C.A.T. created the figures with input from S.D., X.H. and J.M. R.C.A.T., D.T., and S.B. wrote the manuscript with contributions from all authors.

## Competing interests

S.B. is a founder and shareholder of biomodal Ltd, GenomeTx and RNAvate Ltd and a science partner and paid adviser to Ahren Innovation Capital LLC. J.M., M.T., P.G., A.C-U., and W.G. are employees of biomodal Ltd. The remaining authors declare no competing interests.
