## [Transparent Peer Review file · Nature Communications]

Sequencing DNA methylation and hydroxymethylation at co-occurring chromatin features

Corresponding Author: Professor Shankar Balasubramanian

Version 0:

Reviewer comments:

Reviewer #1

(Remarks to the Author)

Epigenetic modifications on DNA and histones act in concert to contribute and reflect changes in chromatin states. In this manuscript, Tavares et al. aimed to integrate the enrichment of histone modifications using the CUT&Tag method with their previously developed 6-letter sequencing, which enables simultaneous detection of 5-methylcytosine (5mC) and 5-hydroxymethylcytosine (5hmC) at single-base resolution.

While detection of 5mC using ChIP or CUT&Tag combined with bisulfite sequencing has been previously reported, the 6-letter sequencing approach represents a novel and important advance. It enables simultaneous detection of both major DNA modifications without bisulfite treatment, thus offering a powerful tool to dissect coordinated transitions between epigenetic states.

Combining CUT&Tag with 6-letter sequencing is not as straightforward as with bisulfite-based methods. To address this, the authors developed a custom hairpin transposase adapter and optimized the removal of DNA fragments lacking adapters on both ends. When comparing the performance of individual protocols with the combined 6-letter CUT&Tag (6L-C&T), they observed that both enrichment and DNA modification detection were retained at comparable efficiency.

The authors then applied 6L-C&T in mouse embryonic stem cells, using antibodies against H3K27ac, H3K4me3, H3K4me1, and H3K27me3. Enrichment revealed changes in DNA methylation and hydroxymethylation when compared to unenriched DNA, consistent with expected chromatin states. Notably, 5mC retention was observed at enhancers marked by H3K4me1, suggesting coordinated regulation at these loci. Furthermore, a machine learning model trained on 6L-C&T data from H3K4me1-marked regions outperformed whole-genome 6-letter sequencing in predicting enhancer states.

In summary, 6L-C&T is a novel technique that offers unique advantages for studying the interplay between DNA and histone modifications. It enables simultaneous, base-resolution profiling of two major DNA modifications on the same DNA molecules and benefits from increased sensitivity due to the absence of bisulfite treatment. The manuscript is well written and suitable for publication. I have following minor suggestions:

1. The authors could discuss how 6L-C&T compares to conventional CUT&Tag in terms of cell input requirements. Does the incorporation of the hairpin adapter and nuclease treatment lead to a significant loss of DNA?

2. It would be interesting to see locus-specific examples of differential methylation (e.g., at imprinted regions such as H19/Igf2), as these would highlight the method's potential to resolve coupling of allele-specific epigenetic states.

(Remarks on code availability)

Reviewer #2

(Remarks to the Author)

Authors report 6-letter CUT&Tag, a multi-omics method for six-base sequencing (GATC/5mC/5hmC) at antibody targeted chromatin features. Approach used in E14 mESCs to profile the co-occurrence of 5mC and 5hmC at a range of histone PTMs (H3K4me1, H3K4me3, H3K27ac, H3K27me3) and relate to the functional state of enhancer elements (poised, primed,

or active). In this example, successful application could offer a more effective means to classify a central transcriptional regulatory element. A potentially valuable approach but extensive revisions would be required to address numerous concerns.

General comments

Much essential background and context are missing, including: how is 5hmC created (e.g., TET enzyme identity, preference and kinetics); any proposed relationships between 5mC and 5hmC (including their relative abundance in cell types that includes mESCs); how this has been measured in prior studies (and how do the authors numbers compare); even the existence of 'third generation' sequencing approaches (ONT, PacBio) and their direct reading of 5mC / 5hmC (whole genome or immunotethered : an extremely active area of study. Such a level of omission on so many essential points is glaring.

Direct Comments & Questions (in no particular order)

'Sequencing-based methods have shaped our understanding of the interplay between these epigenetic marks [5mC and 5hmC]' . What are the current thoughts on this [partic on function gained by 5hmC vs. lost when 5mC is oxidised] ? And outstanding questions that could be addressed using 6L-CUT&Tag?

Scar-filtering is an innovative approach : congratulations to the authors

Datasets exist for native 5mC/5hmC in a variety of sample types (including mESCs). The authors neither mention the alternate approaches used to create same (!), nor make any technical (sensitivity / potential biases) or data comparisons.

Fig 1F: How effective is the '6-letter' approach at distinguishing 5mC / 5hmC on gDNA (e.g., relative efficiency ? Sequence bias (? a 5hmC oligo would not fully inform on same)). How does approach used differ to that in Fig 4c / [PMID: 36747096] (and how has the improvement been achieved) ?

Suppl Fig S1 is a flowchart of 6L-seq library preparation. This would benefit from a more detailed explanation of the key enzymes / steps. As one example, what is the role of DNMT5 ? (also source of this enzyme? Why was it chosen). It should not be necessary to return to the authors prior work [PMID: 36747096] for this background.

Essential: CUT&Tag (and by extension 6L-CUT&Tag) requires an IgG background control for best practice. Whole genome 6L provides a very useful landscape (e.g., does heterogeneity exists at a genomic location?) but cannot identify the % signal at each location that is target-antibody mediated vs. approach background (e.g., as in figs 1D, 1E and 2A). Particularly matters here since 'open' locations (enhancers) are where most of the ATAC-like background can appear in a poorly controlled CUT&Tag experiment (of note H3K27me3 is known to 'dampen' this background, likely because of its sheer abundance).

The central biology explored in the manuscript is the 5mC/5hmC state at mouse enhancers classed as poised, primed, or active (by Cruz-Molina et al) : "we analysed a set of mESC enhancers that have been annotated by the presence or absence of specific histone marks". What justifies this annotated set as a gold standard? It was created by ChIP-seq (a noisier method than CUT&Tag). Has it has been independently verified / widely accepted?

> #187 [& fig 2a whisker plots]: The abundance of 5mC / 5hmC ranges dramatically across the enhancer locations .. which would give this reviewer pause before interpreting biology in differences at the bottom end of the abundance spectrum.

Fig 2a: Each IGV snapshot group should show the distribution of all histone PTMs under study (and an IgG control) : not just those proposed associated with each enhancer class. Windows are too narrow to demonstrate enrichment of H3K4me1/H3K27ac [relative to ? Would need to show flanking locations]. What is driving the scale choice in these locations for %5mC ? [e.g., 1-78 on far left] ?? As below: selected windows are great to make a point but should always be accompanied by whole genome row-linked ChAsE heatmaps).

The authors make extravagant use of violin plots to compare cytosine (hydroxy)methyl enrichment between 6L-CUT&Tag datasets. In many cases the large majority of the distribution is at or near zero, with a thin tail trailing upward : what percentage of the data is in the difference ? e.g., Extended Fig 4b what statistical tests were performed to confirm significance between each distribution?

Extended Data Fig 3a. Authors' choice to use volcano plots (to summarize the enrichment/depletion of cytosine methylation in 6L-CUT&Tag vs 6L-WGS) is non-intuitive. Could a simpler visualization be a dot plot (e.g., X-axis 5mC enrichment in 6L-WGS / Y-axis 5mC enrichment in 6L-CUT&Tag)? Expectation would be that a steep incline supports a greater enrichment of 5mC associated with the histone PTM (as by 6L-CUT&Tag) vs. general landscape (as by 6L-WGS).

Extended Data Fig 6: Used to substantiate "that regions annotated as primed enhancers are enriched with the H3K4me1 mark but also show appreciable levels of H3K27ac and H3K27me3."

Profile plots seem to support that point, since the y-axes indicate more H3K4me1 than H3K27me3 or H3K27ac at these primed enhancers. Please also provide genomic heatmaps showing the three histone PTMs (row-linked; sorted descending by H3K4me1 signal) along with an IgG control (as above) to identify background signals.

(Remarks on code availability)

Reviewer #3

(Remarks to the Author)

(Remarks on code availability)

Reviewer #4

(Remarks to the Author)

This manuscript by Tavares et al. is a study of the 5mC and 5hmC status of CpGs at pulled down histone modification sites in the genome. The authors use their previously developed 6-letter bisulfite free sequencing approach with CUT&Tag. The main advance is the use of the new 6-letter bisulfite free approach, over previous versions of this experiment that have been conducted using bisulfite sequencing. While this is an interesting advance, I do have some concerns that will need to be addressed.

1. There is a lack of data comparison to the previous bisulfite methods, did they see similar things (or not, which is equally interesting).

2. On page 4, the authors state "6L-C&T-derived 5mC and 5hmC levels at each histone mark were lower than the whole-genome average (Extended Data Fig. 1b)", which is really interesting.

However, they later say "active chromatin marks (H3K4me3, H3K27ac and H3K4me1) showed similar numbers of CpGs with higher 5hmC levels than in the whole-genome data and CpGs with lower 5hmC levels than in whole genome" and "the repressive mark (H3K27me3) showed the majority of CpG sites with higher 5hmC in 6L-C&T than in whole-genome"

These two sets of comments seem to be contradictory. The first that all 5mC/5hmC levels were below WG. Then the second two comments that some 5hmC level were similar to WG and some higher than WG. It didn't help that the second two sentences didn't have a figure attached to them.

3. A general trend throughout is that the 5hmC levels always seem to be a very similar % of the 5mC levels (~25-35%). I'm surprised this isn't noted and explained. Is this a factor of how the experiments were carried out? Is this identifying something important about 5hmC formation/removal? Is this the same in the WG samples?

4. I don't understand the point of the machine learning model that was developed, perhaps I misunderstand it. It seems they put the WG 5mC/5hmC and H3K4me1 5mC/5hmC (enhancer marker) data into the ML system and then show that the H3K4me1 version can predict enhancer regions better. But surely that is obvious, as you have preenriched the enhancer regions already?

5. Most of the data is not in the main text, I think it would be useful to have a table(s) to summarise the findings as there is a tremendous amount of data that is skimmed over in the text.

6. In plots Fig 2b, Ext data Fig 4, and Ext data Fig 6b, the violin plots show a very interesting trend. It's almost as if the %5mC/5hmC is quantised. There are specific bumps of data at regular intervals. The authors should explain why this is, as I'm assuming this cannot be biological so will be a factor of how their experiments were conducted or data processed? What effect does this have on the conclusions?

(Remarks on code availability)

Version 1:

Reviewer comments:

Reviewer #1

(Remarks to the Author)

Authors addressed concerns raised in the initial submission of the manuscript, with following remaining points:

1. In imprinted gene loci authors demonstrated that e.g. 6B-C&T for H3K4me3 enriches for low 5mC loci, despite close to 50% 5mC present in WG dataset. These are predicted to be the most dramatic examples in the genome, where half of the alleles exist in two completely distinct chromatin and DNA methylation states. While authors decided not to put this figure in the manuscript, I suggest that they should highlight imprinted genes in volcano plots (for at least H3K4me3).

2. Additional introduction is welcome, but it has a factual error. Statement "5mC is the most abundant cytosine modification (~60% of all Cs in ESCs)9 and 5hmC is the next most abundant cytosine modification (less than 5% of all Cs in ESCs)9 ..." is incorrect. 5mC constitutes close to 2 - 4% of all Cs. Authors likely meant that 60% of Cs in CpG dinucleotides are

methylated, which is closer the actual methylation value. The paper that was referenced does measure modification abundance in CpGs since they perform TLC on labelled nucleotides generated from DNA digestion with MspI, HpaII or Taqα1 – all of them are employed for labelling of Cs in CpGs. Tahiliani et al incorrectly labelled axis in some of their figures, which translated to confusion here.

3. Citations - while listing methods they should also list TAPS (PMID: 30804537) and CAPS (PMID: 33504799) as they are conceptually different from methods that relay on deamination of unmodified Cs (like bisulfite or EM-seq).

(Remarks on code availability)

Reviewer #2

(Remarks to the Author)

General Comments [Co-reviewers #2/3]

The authors have thoroughly addressed our original critiques by an improved introduction (with additional background and technology context), addition of essential IgG control reactions, and additional analyses / figures (including reduced claims of enhancer biology in areas of low PTM signal). The rebuttal is a thorough, good-faith effort to address concerns and describe the manuscript revisions. As such we support publication.

(Remarks on code availability)

Reviewer #3

(Remarks to the Author)

(Remarks on code availability)

Reviewer #4

(Remarks to the Author)

I am happy my comments have been addressed and supportive of publication

(Remarks on code availability)

Note from the authors: the method has now been renamed to 6-base-CUT&Tag (6B-C&T) since the term “base” is preferable to “letter” when referring to nucleobases. All instances of the text and figures, including the original reviewer’s comments below were updated for consistency and to avoid confusion.

Reviewer #1 (Remarks to the Author):

Epigenetic modifications on DNA and histones act in concert to contribute and reflect changes in chromatin states. In this manuscript, Tavares et al. aimed to integrate the enrichment of histone modifications using the CUT&Tag method with their previously developed 6-base sequencing, which enables simultaneous detection of 5-methylcytosine (5mC) and 5-hydroxymethylcytosine (5hmC) at single-base resolution.

While detection of 5mC using ChIP or CUT&Tag combined with bisulfite sequencing has been previously reported, the 6-base sequencing approach represents a novel and important advance. It enables simultaneous detection of both major DNA modifications without bisulfite treatment, thus offering a powerful tool to dissect coordinated transitions between epigenetic states.

Combining CUT&Tag with 6-base sequencing is not as straightforward as with bisulfite-based methods. To address this, the authors developed a custom hairpin transposase adapter and optimized the removal of DNA fragments lacking adapters on both ends. When comparing the performance of individual protocols with the combined 6-base CUT&Tag (6B-C&T), they observed that both enrichment and DNA modification detection were retained at comparable efficiency.

The authors then applied 6B-C&T in mouse embryonic stem cells, using antibodies against H3K27ac, H3K4me3, H3K4me1, and H3K27me3. Enrichment revealed changes in DNA methylation and hydroxymethylation when compared to unenriched DNA, consistent with expected chromatin states. Notably, 5mC retention was observed at enhancers marked by H3K4me1, suggesting coordinated regulation at these loci. Furthermore, a machine learning model trained on 6B-C&T data from H3K4me1-marked regions outperformed whole-genome 6-base sequencing in predicting enhancer states.

In summary, 6B-C&T is a novel technique that offers unique advantages for studying the interplay between DNA and histone modifications. It enables simultaneous, base-resolution profiling of two major DNA modifications on the same DNA molecules and benefits from increased sensitivity due to the absence of bisulfite treatment. The manuscript is well written and suitable for publication. I have following minor suggestions:

1. The authors could discuss how 6B-C&T compares to conventional CUT&Tag in terms of cell input requirements. Does the incorporation of the hairpin adapter and nuclease treatment lead to a significant loss of DNA?

In conventional CUT&Tag, theoretically half of all tagmented fragments are not amplified as they have the same adaptor at both ends (e.g., A/A and B/B), whereas fragments with A/B and B/A adaptor combinations are successfully amplified in PCR. In 6B-C&T, insertion of the same hairpin adaptor at either end of the DNA fragment means that in principle all double-sided tagmentation events (both ends of the fragment) should be recovered by PCR.

In practice, we have found that most fragments generated in 6B-C&T lack an adaptor at one or both ends. These fragments probably originate from DNA shearing or non-specific nuclease activity throughout the protocol and are efficiently repaired, ligated and amplified in the 6-base-seq workflow. Importantly, we developed a method to minimize the number of these fragments via exonuclease treatment and to computationally exclude them from downstream analysis. The increased number of steps and processing in the 6B-seq protocol does result in lower DNA yield relative to conventional CUT&Tag. However, we compensate for this by scaling up the starting number of cells four-fold, which results in library yields and read duplication rates similar to conventional CUT&Tag.

The Methods now include a statement justifying the input requirements of 6B-C&T in comparison to conventional CUT&Tag:

“Since the size selection and 6-base enzymatic conversion steps reduce DNA recovery, we increased input material to maintain DNA yields and PCR duplication rates comparable to conventional CUT&Tag. Four CUT&Tag samples (500,000 cells each) were prepared in parallel per biological sample.”

2. It would be interesting to see locus-specific examples of differential methylation (e.g., at imprinted regions such as H19/Igf2), as these would highlight the method's potential to resolve coupling of allele-specific epigenetic states.

We examined differential methylation at several genes reported to show allele-specific expression during mouse ESC differentiation (PMID: 30767785, PMID: 34226608). Please see below (Review-only Figure 1) for specific examples.

The *Peg3/Usp29* bidirectional promoter exemplifies a general epigenetic pattern and harbours an imprinting control region (PMID: 27257070). It is characterized by a CpG-rich domain with global 5mC levels approximately half those of the surrounding regions, as determined by whole-genome 6-base sequencing, and by a pronounced enrichment of the promoter-associated histone mark (H3K4me3). This gene is paternally expressed in ESCs, with the paternal (active) allele associated with H3K4me3 and the maternal (repressed) allele associated with H3K9me3 (PMID: 34226608). Accordingly, %5mC at H3K4me3 (given by H3K4me3 6B-C&T) is ~0, indicating this histone mark is coupled with the unmethylated allele. No coupling could be determined with the other histone marks we profiled (H3K27me3, H3K4me1, H3K27ac), as they are not significantly enriched at this locus. Loci such as *Kcnq1ot1* and *Igf2r/Airn* display similar patterns. *H19* does not show appreciable levels of the histone marks profiled in our study, and we did not detect differential methylation at *Igf2* in our data.

[editorial note: review-only figure redacted

6B-C&T is ideally suited to detect allele-specific coupling of histone marks with 5mC and 5hmC. We anticipate that exploratory analysis of histone modifications such as H3K9me3 and H3K20me3 will be highly informative in this context, since these marks are strongly associated with differential methylation at imprinted loci (PMID: 20180964).

Reviewer #2 (Remarks to the Author):

Authors report 6-base CUT&Tag, a multi-omics method for six-base sequencing (GATC/5mC/5hmC) at antibody targeted chromatin features. Approach used in E14 mESCs to profile the co-occurrence of 5mC and 5hmC at a range of histone PTMs (H3K4me1, H3K4me3, H3K27ac, H3K27me3) and relate to the functional state of enhancer elements (poised, primed, or active). In this example, successful application could offer a more effective means to classify a central transcriptional regulatory element. A potentially valuable approach but extensive revisions would be required to address numerous concerns.

General comments

1. Much essential background and context are missing, including: how is 5hmC created (e.g., TET enzyme identity, preference and kinetics); any proposed relationships between 5mC and 5hmC (including their relative abundance in cell types that includes mESCs); how this has been measured in prior studies (and how do the authors numbers compare); even the existence of 'third generation' sequencing approaches (ONT, PacBio) and their direct reading of 5mC / 5hmC (whole genome or immunotethered : an extremely active area of study. Such a level of omission on so many essential points is glaring.

Additional background has been added (see pages 1-2 – Intro paragraphs 1-2), including the relative 5mC/5hmC abundance in mESCs:

“Chemical modifications of DNA bases and histone tails regulate physical and functional properties of chromatin, including DNA accessibility, genomic architecture and binding of transcription regulators¹⁻⁴. Methylation of cytosine bases on DNA is critical to establish the epigenetic landscape and governs gene activity and inheritance⁵. Cytosine bases can be methylated by DNA methyltransferases (DNMTs), that transfer a methyl group from the donor S-adenosylmethionine to form 5-methylcytosine (5mC)⁶, which can be further oxidised to form 5-hydroxymethylcytosine (5hmC) through the action of the Ten-Eleven Translocation family of proteins (TETs 1-3)⁷. TET enzymes can further oxidise 5hmC to other intermediates like 5-formylcytosine (5fC) and 5-carboxylcytosine (5caC)⁸, each of which can be excised by thymine DNA glycosylase, followed by restoration of C, completing a demethylation cycle (Figure 1). 5mC is the most abundant cytosine modification (~60% of all Cs in ESCs)⁹ and 5hmC is the next most abundant cytosine modification (less than 5% of all Cs in ESCs)⁹, with 5fC and 5caC being present at substantially lower levels (~0.002% and ~0.0003%, respectively)⁸. While 5hmC can be an intermediate in active DNA demethylation, it has also been shown to be a stable epigenetic mark that is associated with transcriptional activity and cell fate¹⁰⁻¹⁵. The functional roles of 5mC and 5hmC depend on their genomic context. Broadly, 5mC enrichment at promoters and enhancers is associated with transcriptional repression^{16, 17}, whereas 5hmC is enriched at active regulatory regions – particularly enhancers – and also within the gene bodies of transcriptionally active genes^{18, 19}”

“The development of sequencing methods to detect 5mC and 5hmC has been an important focus in epigenomics. While early methods such as bisulfite sequencing²⁰ cannot distinguish 5hmC from 5mC, later approaches – including oxidative bisulfite sequencing²¹, TET-assisted bisulfite sequencing²², APOBEC-coupled epigenetic sequencing²³ and third-generation sequencing platforms (e.g., PacBio, Oxford Nanopore)²⁴ – have enabled base-resolution mapping of both 5mC and 5hmC. More recent methods have now leveraged state-of-the-art enzymology and chemistry to enable bisulfite-free and simultaneous detection of 5mC and 5hmC on the same DNA fragment²⁵⁻²⁷.”

Our data is consistent with previous reports showing the relative genome-wide abundance of 5mC corresponds to ~10-30X that of 5hmC. This information is now included in the text (page 3, Results paragraph 4):

“As a reference, we also performed whole-genome 6-base-seq, which provides an untargeted picture representative of the total pool of DNA fragments. Using this approach, we confirmed that the relative 5mC/5hmC abundance was consistent with reported values in mESCs (Extended Data Fig. 1a).”

Direct Comments & Questions (in no particular order)

2. 'Sequencing-based methods have shaped our understanding of the interplay between these epigenetic marks [5mC and 5hmC]' . What are the current thoughts on this [partic on function gained by 5hmC vs. lost when 5mC is oxidised] ? And outstanding questions that could be addressed using 6B-

CUT&Tag?

See response to Referee 2 point 1 above together with the following introductory text (page 2, Intro paragraph 2):

“Despite such advancements, a key challenge in the field has been to determine how 5mC and 5hmC are coupled in space and time to other chromatin features such as histone marks, transcription factors and chromatin remodelers. Establishing this relationship is critical to appreciate how epigenetic features are coupled, when and where they co-occur, and how they determine causal relationships and regulatory roles. Current strategies that map DNA modifications and chromatin features largely rely on comparing profiling data from independent experiments for different features. This approach does not necessarily determine co-occurrence of features at the molecular level (Fig. 1b).”

Scar-filtering is an innovative approach : congratulations to the authors

3. Datasets exist for native 5mC/5hmC in a variety of sample types (including mESCs). The authors neither mention the alternate approaches used to create same (!), nor make any technical (sensitivity / potential biases) or data comparisons.
 - 3.1 With respect the citation of approaches to measure 5hmC/5mC see response to Referee 2 point 1.
 - 3.2 To our knowledge, 6B-C&T is the first method to directly resolve 5mC and 5hmC bases on histone mark-containing DNA fragments. Previous 6-base sequencing methods have so far been performed solely on purified DNA for whole genome samples (e.g., PMID: 38846078, PMID: 38336903). To understand the main technical differences between the whole-genome vs targeted approaches, we have also performed extensive comparisons between whole-genome 6-base-seq and 6B-C&T throughout the manuscript (Extended Data Figs. 1-3).
 - 3.3 Our findings agree with reports that used other methods for simultaneous detection of DNA methylation at histone marks like ChIP-BS (PMID: 33844685), CUT&Tag-BS (PMID: 35028637) and scEpi-seq (PMID: 40999097). Specifically, we observe similar levels of total DNA methylation at the histone marks profiled in common with those studies, such as H3K4me1 and H3K27me3. In addition, our detection of different histone mark-associated fragments mapping to primed enhancers confirms and expands earlier characterisations of epigenetic heterogeneity at enhancer loci (PMID: 33844685).

The revised Discussion highlights the consistency between these previous reports and our study (page 5, Discussion paragraph 1):

“6B-C&T detects 5mC and 5hmC bases that co-occur with target chromatin features. Profiling of major regulatory histone modifications, such as H3K4me1 and H3K27me3, reveals patterns that are consistent with other

measurements of total DNA methylation at these marks^{31, 39, 40} and adds 5hmC as a new dimension to the multiomics analysis of individual DNA fragments.”

We have also directly compared our 6B-C&T data to publicly available data profiling total DNA methylation (modC) at H3K4me1 and H3K27me3 via CUT&Tag-BS (PMID: 35028637). We observed good agreement between the two methods both at the level of CUT&Tag fragment enrichments and at the level of modC detection (Supplementary Figure 6a-b, shown below). We also observed excellent agreement of total DNA methylation data from both methods at the enhancer regions studied throughout our manuscript (Supplementary Figure 6c).

Results have been updated to include the above comparisons (page 3, Results paragraph 4; page 4, Results paragraph 7):

“...we assessed how well 6B-C&T correlates with CUT&Tag-BS, a method that profiles total cytosine methylation (modC) on CUT&Tag DNA fragments via bisulfite conversion. Both genomic feature enrichment and CpG modification levels for 6B-C&T agreed well with publicly available CUT&Tag-BS data for H3K4me1 and H3K27me3 (Pearson $r \geq 0.83$, Supplementary Fig. 6).”

“Excellent agreement was also obtained for total DNA methylation levels (%modC) at all enhancer types between 6B-C&T and available C&T-BS data for the general enhancer mark, H3K4me1 (Supplementary Fig. 6c).”

This confirms 6B-C&T can retrieve published patterns of simultaneous histone mark and DNA methylation accumulation in addition to deconvolving 5mC from 5hmC on the same DNA fragment.

Supplementary Figure 6. Comparison between 6-base-CUT&Tag (this study) and available CUT&Tag-BS (GSE179266) data. (a) Pearson correlation matrices of genomic fragment enrichments for 6B-C&T compared to a publicly available C&T-BS dataset for two histone marks: H3K4me1 (left) and H3K27me3 (right). **(b)** Correlation matrices for genome-wide total DNA methylation (modC) fractions for the same datasets (left, H3K4me1; right, H3K27me3). Histograms of modified CpG fractions for 200-bp genomic tiles from each available experimental replicate are shown on the diagonal alongside scatter plots below the diagonal. Pearson correlation coefficients are displayed for each pairwise comparison above the diagonal. **(c)** Comparison of total DNA methylation (modC) from H3K4me1 6B-C&T and H3K4me1C&T-BS at enhancers. Stacked bar plots show % modC (green) and % C (grey) for different sets of annotated enhancer regions. Error bars represent standard deviation (3 independent experiments shown for 6B-C&T; 2 independent experiments shown for C&T-BS).

4. Fig 1F: How effective is the '6-base' approach at distinguishing 5mC / 5hmC on gDNA (e.g., relative efficiency ? Sequence bias (? a 5hmC oligo would not fully inform on same)). How does approach used differ to that in Fig 4c / [PMID: 36747096] (and how has the improvement been achieved) ?

Fullgrabe et al. (PMID: 36747096) have described how the 6-base method has comparable/superior detection sensitivity and specificity when measured against other platforms, including bisulfite-based and enzymatic methods. It is important to note that 6-base-seq can only distinguish 5hmC from 5mC at CpG sites. Methylation outside of the CpG sequence context, such as CHH, is detected but cannot be deconvolved due to the method's enzymology. Given the majority of (hydroxy)methylation in mESCs is present at CpG sites (PMID: 25042786), this was not a limitation in our study.

While the ground-truth spike-in controls provide an accurate estimate of the method's detection sensitivity and specificity, we also assessed whether 5hmC detection exhibits significant sequence preferences beyond the CpG context. Using chromosome 1 as an example (see Review-only Figure 2 below), we performed motif enrichment analysis using HOMER on 5hmC sites identified in 6-base-seq. The analysis revealed multiple distinct sequence motifs present at genomic 5hmC sites, which indicates that 5hmC is detected across diverse genomic contexts with no strong representation of a specific sequence bias other than the CpG context.

[editorial note: review-only figure redacted]

Regarding how our 6-base-seq approach compares to Fullgrabe et al.: the main difference is that the previous study used sonication to shear gDNA, followed by end repair and hairpin ligation (3 steps), whereas we use Tn5 tagmentation to

simultaneously fragment and insert the hairpin adaptor (1 step). In our approach, we observe improved efficiency of 6-base-seq library generation, which is accompanied by slightly improved 5mC and 5hmC detection (Fig. 2d) when compared to the results shown in Fig 4c of Fullgrabe et al. (PMID: 36747096).

5. Suppl Fig S1 is a flowchart of 6B-seq library preparation. This would benefit from a more detailed explanation of the key enzymes / steps. As one example, what is the role of DNMT5 ? (also source of this enzyme? Why was it chosen). It should not be necessary to return to the authors prior work [PMID: 36747096] for this background.

We have now explained the workflow in detail (Supplementary Figure 1a, caption pasted below for convenience). We also reference the original 6-base-seq paper (Fullgrabe et al. 2023, PMID: 36747096), where extensive details are provided, including the role of DNMT5, chosen for its high specificity for copy methylation over de novo methylation. Details of all reagents used for 6-base-seq library preparation, which are commercially available, are also provided in the Methods of the present manuscript.

“Supplementary Fig. 1. Detailed 6-base-seq library preparation from tagged DNA. (a) Workflow including DNA circularization and exonuclease treatment. For clarity, subsequent enzymatic steps are shown only for the bottom strand of the initial dsDNA fragment. DNA is first circularised by gap filling with Phusion DNA polymerase and nicks sealed with Taq DNA ligase. Subsequent exonuclease treatment specifically digests linear DNA while dumbbell-like fragments are protected. (Uracil-specific excision reagent) USER cleavage then generates a single nucleotide gap at both uracil sites present in each hairpin. This allows for strand separation of the original duplex via heat denaturation and hairpin re-folding on each strand via slow cooling. Klenow (3' → 5' exo-) next synthesises a copy strand on each individual strand. T4 polynucleotide kinase (PNK) is included for gap repair of USER-digested sites. A “fork-head” adaptor is then ligated to the free end of double-strand DNA fragments. DNMT5, chosen for its high specificity of copy methylation over de novo methylation, then specifically methylates the CpG on the recently synthesised copy strand only when the original strand contains a 5mCpG1. TET2 is next used to oxidise all 5mC groups to 5hmC followed by their protection in the same reaction by glycosylation using beta-glucosyltransferase (βGT). Finally, APOBEC3A (ssDNA deaminase) converts all unprotected cytosines to uracils, assisted by the dsDNA unwinding activity of the UvrD helicase. Primers complementary to the fork-head adaptor sequences and containing Illumina P5 and P7 sequences are then used to amplify all fragments by PCR to generate a library for Illumina sequencing. All cytosine states are schematically represented and colour-coded according to the legend. “5mC” = 5-methylcytosine; “5hmC” = 5-hydroxymethylcytosine; “5ghmC” = 5-glucosylhydroxymethylcytosine.”

6. Essential: CUT&Tag (and by extension 6B-CUT&Tag) requires an IgG background control for best practice. Whole genome 6B provides a very useful landscape (e.g., does heterogeneity exist at a genomic location?) but cannot identify the % signal at each location that is target-antibody mediated vs. approach background (e.g., as in figs 1D, 1E and 2A). Particularly matters here since 'open' locations (enhancers) are where most of the ATAC-like

Response to reviewers

background can appear in a poorly controlled CUT&Tag experiment (of note H3K27me3 is known to 'dampen' this background, likely because of its sheer abundance).

We have now performed 6B-C&T on E14TG2A mESCs using a non-specific IgG control to assess the amount of accessibility-driven background that is unrelated to target recognition by a specific antibody.

To estimate the contribution of non-specific background to the total signal detected with 6B-C&T genome-wide, we compared the percentage of USER-scarred read pairs for histone mark-targeted libraries, which are a proxy for total signal, and IgG control libraries, which serve as a proxy for the background:

N_s = number of reads with scars on both ends (bona fide tagmentation events) and N_{ns} = number of reads with no scars or with single-end scars (random fragmentation, DNA shearing, partial tagmentation).

$$F_{scars} \text{ (fraction of scarred pairs)} = \frac{N_s}{N_s + N_{ns}} = \frac{1}{1 + \frac{N_{ns}}{N_s}}$$

$$\rightarrow \frac{N_s}{N_{ns}} = \frac{F_{scars}}{(1 - F_{scars})} = \text{(scarred/unscarred fragment ratio)}.$$

We can use the scarred/unscarred fragment ratio to estimate the contribution of non-specific, accessibility-driven background (given by IgG) to the total signal observed in the histone mark-targeted experiment:

$$\frac{\frac{N_s}{N_{ns}(\text{IgG control})}}{\frac{N_s}{N_{ns}(\text{histone mark})}} = \frac{\frac{F_{scars}(\text{IgG control})}{[1 - F_{scars}(\text{IgG control})]}}{\frac{F_{scars}(\text{histone mark})}{[1 - F_{scars}(\text{histone mark})]}}$$

On average, F_{scars} for IgG libraries = 1.6% and F_{scars} for histone mark libraries ~ 55% (range = 50-60%):

$$\frac{\frac{N_s}{N_{ns}(\text{IgG control})}}{\frac{N_s}{N_{ns}(\text{histone mark})}} = \frac{\frac{0.016}{[1 - 0.016]}}{\frac{0.55}{[1 - 0.55]}} = 0.013, \text{ or } 1.3\%$$

From the above figure, the IgG-estimated background represents, on average, ~1-2% of the total signal detected by histone mark 6B-C&T, which confirms the high specificity of this method.

Also see responses to points 9 and 12, which illustrate the above at specific loci and with genome-wide heatmaps.

7. The central biology explored in the manuscript is the 5mC/5hmC state at mouse enhancers classed as poised, primed, or active (by Cruz-Molina et al) : “we analysed a set of mESC enhancers that have been annotated by the presence or absence of specific histone marks”. What justifies this annotated

set as a gold standard? It was created by ChIP-seq (a noisier method than CUT&Tag). Has it has been independently verified / widely accepted?

We chose this annotation set as it was comprehensive and derived independently from CUT&Tag. This dataset covered the same histone modifications that we investigated using the same E14TG2A mESC cells and importantly the same culture conditions (serum/LIF medium) and so represents the same cell state of pluripotency/differentiation. Furthermore, a key feature in this dataset is the use of P300 occupancy as a criterion for enhancer type assignment, which enables the active enhancers to be confidently distinguished from other classes. This dataset has been widely used as a reference in other studies of mESCs (see PMID: 33620319, PMID: 32115407, PMID: 37456851, PMID: 40743391, PMID: 34792172). While we could have annotated enhancers *de novo* using our CUT&Tag data, we felt that it was essential to use comparative data independent of Tn5-based methods and their associated sequence/regional biases. This choice enabled us to cross-validate our genomic assignments against previously published, orthogonal data.

Supporting this choice of enhancer annotation, we observed good agreement between 6B-C&T and the ChIP-seq data from Cruz-Molina et al. for all enhancer-associated histone marks at the reference loci used in our study (Supplementary Fig. 9, see below). This is now mentioned in the Results (page 4, Results paragraph 7) and rules out the noisier character of ChIP-seq as a potential confounder in the present study.

“We observed good agreement at these loci between 6B-C&T and the ChIP-seq data originally used for their annotation (Supplementary Fig. 9).”

Supplementary Fig. 9. Comparison between 6-base-CUT&Tag (this study) and ChIP-seq data used for enhancer class annotation (GSE89211). Correlation scatter plots for ChIP-seq data from Cruz-Molina et al. (2017) and 6B-C&T data for enhancer-associated histone marks are plotted for different sets of annotated enhancer regions: **(a)** active enhancers (H3K27ac and H3K4me1), **(b)** primed enhancers (H3K4me1) and **(c)** poised enhancers (H3K27me3 and H3K4me1). In all cases, CPM-normalised genomic enrichments (bin size = 10 bp) are plotted for each experiment (y-axis = ChIP-seq and x-axis = 6B-C&T) and R-squared (R^2) values are annotated on each plot. Least squares polynomial (first-degree) fitting was performed with *numpy.polyfit* to produce the fitting line. A single experimental replicate from each experiment was used as representative of the overall trend. **(d)** Pearson correlation matrix for data plotted in **a-c**. Pearson correlation coefficients (Pearson's R) are shown for each pairwise comparison.

8. > #187 [& fig 2a whisker plots]: The abundance of 5mC / 5hmC ranges dramatically across the enhancer locations .. which would give this reviewer pause before interpreting biology in differences at the bottom end of the abundance spectrum.

We observed significantly higher 5mC and 5hmC levels at primed enhancers compared to active and poised enhancers. We note that within each enhancer type, 5mC and 5hmC levels can span a wide range. Coupled with intrinsic heterogeneity across the cell population, lower abundances of these marks will be harder to interpret biologically without other experimental data. We have therefore focused our main analysis on the higher levels of histone mark-specific 5mC and 5hmC found at primed enhancers.

9. Fig 2a: Each IGV snapshot group should show the distribution of all histone PTMs under study (and an IgG control) : not just those proposed associated with each enhancer class. Windows are too narrow to demonstrate enrichment of H3K4me1/H3K27ac [relative to ? Would need to show flanking locations]. What is driving the scale choice in these locations for %5mC ? [e.g., 1-78 on far left] ?? As below: selected windows are great to make a point but should always be accompanied by whole genome row-linked ChAsE heatmaps).

Each IGV snapshot now includes all enhancer histone marks along with the IgG control (see below). Windows were expanded to include flanking regions and the annotated enhancer under study is now indicated in each panel. We note that the poised enhancer example (*Hoxa11* locus) is part of a bigger *Hox* gene cluster (~120 kb) which is entirely covered by H3K27me3 signal in E14TG2A mESCs. Further expanding the window in this case to capture the H3K27me3 signal drop would greatly reduce the amount of detail and interpretability of the snapshot.

The y-axis scales, previously presented as per IGV default, are now adjusted to fit the data range in each window and all tracks within each group now have the same scale (I – CUT&Tag enrichment; II – %5mC and III – %5hmC):

Figure 3a. Genome browser views of representative active, primed and poised enhancer loci. Tracks show histone mark levels and percentages of 5mC and 5hmC obtained with 6B-C&T

for enhancer-associated histone marks, in addition to IgG control. The genomic location of each annotated enhancer is indicated with a horizontal bar and shaded area coloured according to enhancer type (active – blue, primed – green and poised – red). Tracks within each group (CUT&Tag fragment enrichments, % 5mC and % 5hmC) have the same y-axis scale.

Boxplot quantification of the full windows on IGV snapshots above and of the annotated enhancers are now shown in Supplementary Figure 10:

Supplementary Fig. 10 Boxplots of %5mC and %5hmC in genomic windows (IGV snapshots) shown in Main Figure 3. (a) Boxplots of %5mC from 6B-C&T at enhancer marks (H3K27me3 in red, H3K27ac in blue and H3K4me1 in green) and IgG control (black) for active (left column), primed (centre column) and poised (right column) enhancer loci. The top panels show the full genomic windows in Main Figure 3 and the bottom panels show only the

annotated enhancer window in each case. **(b)** Same as in **a** but for %5hmC. Whiskers represent $Q3 + 1.5 \times \text{interquartile range}$ (upper) and $Q1 - 1.5 \times \text{interquartile range}$ (lower).

For genome-wide heatmaps, please see response to point 12.

10. The authors make extravagant use of violin plots to compare cytosine (hydroxy)methyl enrichment between 6B-CUT&Tag datasets. In many cases the large majority of the distribution is at or near zero, with a thin tail trailing upward : what percentage of the data is in the difference ? e.g., Extended Fig 4b what statistical tests were performed to confirm significance between each distribution?

We have now added boxplots to Extended Data Figure 4 to indicate the median, interquartile range, lower and upper bounds of each distribution in the violin plots (see below). In all cases, 80-95% of all data points are found within the distribution ($< 3^{\text{rd}}$ quartile + 1.5X interquartile range). The following were added to Supplementary Table 1: Wilcoxon rank sum test p-values that confirm significance for each comparison; total number of CpGs analysed; upper bound values for each distribution and the percentage of CpGs outside of the distribution (outliers).

Extended Data Figure 4. 5mC and 5hmC levels associated with different histone marks at mESC enhancers. (a) Percentage 5mC and 5hmC at CpGs at active enhancers vs CpGs outside active enhancers (labelled “other CpGs”). H3K27ac 6B-C&T data (blue vs grey) and H3K4me1 6B-C&T data (green vs grey) are shown for this enhancer type. (b) Percentage 5mC and 5hmC at CpGs at primed enhancers vs other CpGs. H3K4me1 6B-C&T data (green vs grey) is plotted for both groups of CpGs in each case. (c) Percentage 5mC and 5hmC at CpGs at poised enhancers vs other CpGs. H3K27me3 6B-C&T data (red vs grey) and H3K4me1 6B-C&T data (green vs grey) are shown for this enhancer type. All distributions are represented with violin plots (light grey) accompanied by boxplots (black) to indicate the median, interquartile range and upper bound values (3rd quartile + 1.5 x interquartile range). Descriptive statistics and Wilcoxon rank sum test results are listed in Supplementary Table 1.

11. Extended Data Fig 3a. Authors’ choice to use volcano plots (to summarize the enrichment/depletion of cytosine methylation in 6B-CUT&Tag vs 6B-WGS) is non-intuitive. Could a simpler visualization be a dot plot (e.g., X-axis 5mC enrichment in 6B-WGS / Y-axis 5mC enrichment in 6B-CUT&Tag)? Expectation would be that a steep incline supports a greater enrichment of 5mC associated with the histone PTM (as by 6B-CUT&Tag) vs. general landscape (as by 6B-WGS).

The scatter plot has now been added as suggested (Supplementary Fig. 8, see below) and closely matches the reviewer’s prediction: sites where 6B-C&T > WG follow a trend closer to the y-axis (steep incline), while sites with 6B-C&T < WG follow a trend closer to the x-axis (flat incline), deviating from the diagonal slope. The volcano plots were kept in Extended Data as they provide a more complete view of differentially (hydroxy)methylated sites by integrating the fold-change and statistical significance for each site.

Supplementary Figure 8. Differentially methylated CpGs (DMCs) and differentially hydroxymethylated CpGs (DHMCs) between 6B-C&T and whole-genome (untargeted) 6-base-seq. Scatter plots of %5mC (top row) and %5hmC (bottom row) for CpG sites (round dots) captured in both 6B-C&T and whole-genome (WG) 6-base-seq. In each plot, the y-axis coordinate of each CpG site is obtained from 6B-C&T data (replicate-averaged) and the x-axis coordinate is obtained from WG data (replicate-averaged). Differentially methylated sites (p -value < 0.05, $|\log_2FC| \geq 0.5$) between both methods are coloured for each histone mark (H3K27ac in blue, H3K4me3 in orange, H3K4me1 in green and H3K27me3 in red). Non-significant changes are shown in grey.

12. Extended Data Fig 6: Used to substantiate “that regions annotated as primed enhancers are enriched with the H3K4me1 mark but also show appreciable levels of H3K27ac and H3K27me3.” Profile plots seem to support that point, since the y-axes indicate more H3K4me1 than H3K27me3 or H3K27ac at these primed enhancers. Please also provide genomic heatmaps showing the three histone PTMs (row-linked; sorted descending by H3K4me1 signal) along with an IgG control (as above) to identify background signals.

We have now added the suggested heatmaps in Extended Data Fig. 5 (see below).

Extended Data Figure 5. Genomic enrichments of 6B-C&T for enhancer-associated histone modifications at annotated enhancer types. Genomic heatmaps for each histone mark (H3K4me1 in green; H3K27ac in blue; H3K27me3 in red and IgG control in grey) are plotted in +/- 6 kilobase windows relative to the centre of each enhancer locus for a set of annotated **(a)** active, **(b)** primed and **(c)** poised enhancers.

Reviewer #3 (Remarks to the Author):

Reviewer #4 (Remarks to the Author):

This manuscript by Tavares et al. is a study of the 5mC and 5hmC status of CpGs at

pulled down histone modification sites in the genome. The authors use their previously developed 6-base bisulfite free sequencing approach with CUT&Tag. The main advance is the use of the new 6-base bisulfite free approach, over previous versions of this experiment that have been conducted using bisulfite sequencing. While this is an interesting advance, I do have some concerns that will need to be addressed.

1. There is a lack of data comparison to the previous bisulfite methods, did they see similar things (or not, which is equally interesting).

See response to reviewer 2, points 1 and 3.

2. On page 4, the authors state “6B-C&T-derived 5mC and 5hmC levels at each histone mark were lower than the whole-genome average (Extended Data Fig. 1b)”, which is really interesting.

However, they later say “active chromatin marks (H3K4me3, H3K27ac and H3K4me1) showed similar numbers of CpGs with higher 5hmC levels than in the whole-genome data and CpGs with lower 5hmC levels than in whole genome” and “the repressive mark (H3K27me3) showed the majority of CpG sites with higher 5hmC in 6B-C&T than in whole-genome”

These two sets of comments seem to be contradictory. The first that all 5mC/5hmC levels were below WG. Then the second two comments that some 5hmC level were similar to WG and some higher than WG. It didn't help that the second two sentences didn't have a figure attached to them.

The first comment compares the **different genomic regions** captured by 6B-C&T vs untargeted (whole-genome) 6-base-seq experiments. 6B-C&T captures a subset of the whole genome, i.e., histone mark-enriched sites. This subset is generally less (hydroxy)methylated than the rest of the genome, which means average 5mC and 5hmC levels obtained from 6B-C&T are lower than in the whole-genome case.

The second comment refers to a focused analysis comparing 6B-C&T data to untargeted (whole-genome) data at the **same (shared) CpG sites**. The observed differences reflect the different DNA fragments captured by the two methods: 6B-C&T fragments are physically associated with each histone mark whereas whole-genome fragments encompass all DNA fragments mapping to the same sites, which includes those not associated with each respective histone mark in the population. This results in the CpG-level differences shown in Extended Data Figure 3.

We have rewritten the relevant Results section (pages 3-4, Results paragraphs 5-6) and now mention each figure next to the relevant text. To remove any ambiguity between the two sets of comments, the CpG-level comparison between 6B-C&T and WG is now phrased in terms of differentially methylated CpGs (DMCs) and differentially hydroxymethylated CpGs (DHMCs). New Supplementary Figure 8 contains an alternative visualisation of DMCs and DHMCs in x-y scatter plots (see response to reviewer 2, point 11).

“Methylation is known to be depleted in histone mark-enriched regulatory regions relative to the other regions of the in mESC genome. Consistent with this, we observed that average 5mC and 5hmC levels in 6B-C&T were lower than in the whole-genome (untargeted) 6-base-seq experiment (Extended Data Fig. 1a). This difference is due to the 6B-C&T method specifically capturing DNA fragments at histone mark-enriched sites, which are generally less (hydroxy)methylated compared to the whole genome.”

“We next investigated how 5mC and 5hmC levels in 6B-C&T differed from untargeted 6-base-seq data for the same CpG sites (Extended Data Fig. 3a). First, we identified CpG sites with statistically significant differences between both methods, i.e., differentially methylated CpGs (DMCs) and differentially hydroxymethylated CpGs (DHMCs). While most DHMCs between both methods were located in introns followed by intergenic and promoter regions, DMCs were primarily found in promoters followed by introns/exons (Extended Data Fig. 3b-e). Overall, DMCs predominantly showed lower 5mC levels in 6B-C&T relative to whole-genome (untargeted) data (Extended Data Fig. 3a, Supplementary Fig. 8). This suggests that DNA associated with histone modifications at these sites is mostly hypomethylated relative to the total pool of DNA fragments. For DHMCs, active chromatin marks (H3K4me3, H3K27ac and H3K4me1) showed similar numbers of sites with higher or lower 5hmC levels in 6B-C&T relative to the untargeted experiment; in contrast, for the repressive mark (H3K27me3) the majority of DHMCs showed higher 5hmC levels in 6B-C&T than in untargeted 6-base-seq (Extended Data Fig. 3a, Supplementary Fig. 8).”

3. A general trend throughout is that the 5hmC levels always seem to be a very similar % of the 5mC levels (~25-35%). I'm surprised this isn't noted and explained. Is this a factor of how the experiments were carried out? Is this identifying something important about 5hmC formation/removal? Is this the same in the WG samples?

Even though the 5hmC/5mC ratio varies over a relatively narrow range as pointed out by the reviewer, it reveals an interesting histone mark-dependent trend that is not captured by WG data. The following plots (see below), now included as Supplementary Figure 7, compare the relative abundances of 5hmC and 5mC from WG and 6B-C&T data at different histone mark sites.

Response to reviewers

Supplementary Figure 7. Relative abundances of 5mC and 5hmC from WG 6-base-seq and 6B-C&T at four histone modifications. In each graph, stacked bar plots show WG and 6B-C&T at CpG sites commonly detected by both experiments. 5mC is represented in dark and 5hmC in light colours. Percentages of each base are annotated for each condition.

5hmC/5mC from 6B-C&T increases in the following order: H3K27me3 (repressed chromatin, 20%/80%) → H3K4me1 (enhancers, 24%/76%) → H3K27ac (active chromatin, 30%/70%) ~ H3K4me3 (promoters, 31%/69%). This suggests that 5mC removal and 5hmC accumulation might be physically coupled with each histone mark in a transcriptional activation-dependent manner. Conversely, 5hmC/5mC from WG is generally invariant (15%/85%), indicating that these individual trends are masked by the average landscape of DNA fragments at the same sites.

In the manuscript, we highlight this histone mark dependence of the 5hmC/5mC ratio in the following section (page 3, Results paragraph 5):

“Furthermore, when looking at individual histone marks, we found that those associated with active chromatin (H3K4me3, H3K27ac, H3K4me1) showed lower overall methylation and a higher 5hmC/5mC ratio than the bivalent/repressed chromatin (H3K27me3) mark, a pattern that was absent in whole-genome data (Extended Data Fig. 1b, Supplementary Fig. 7).”

4. I don't understand the point of the machine learning model that was developed, perhaps I misunderstand it. It seems they put the WG 5mC/5hmC and H3K4me1 5mC/5hmC (enhancer marker) data into the ML system and then show that the

H3K4me1 version can predict enhancer regions better. But surely that is obvious, as you have preenriched the enhancer regions already?

To clarify, the purpose of the ML analysis was to test whether 5mC and 5hmC information for single histone mark (H3K4me1, common to all enhancer types) would robustly predict *different* enhancer states. It is important to mention that although all enhancer regions are marked by H3K4me1, whether 5mC and 5hmC physically co-occur with this histone modification on the same DNA fragment had never been measured. The results show that these two layers of epigenetic information (*i.e.*, *histone and DNA modifications*) are coupled and intrinsically linked to different enhancer functional states. Furthermore, our approach demonstrates how informative methylation signatures can be extracted to define different enhancer states without the need for whole-genome sequencing or multiple histone mark enrichment profiles.

We have clarified the relevant text:

Results (pages 4-5, Results paragraph 9):

“H3K4me1 is common to all enhancer functional states, so we next asked whether 6B-C&T methylation at this single feature better resolves different enhancer types compared to the more heterogenous whole genome (untargeted) 6-base-seq data. For this, we used H3K4me1 6B-C&T data to train a machine learning model for classification of enhancer types (active, primed and poised) and compared its performance to a model with the same architecture trained on whole-genome 6B-seq data (Fig. 4a). For all enhancer types, the 6B-C&T-derived model showed superior performance when compared to the whole genome-derived model (Fig. 4b-d; Extended Data Fig. 7). This demonstrates that the H3K4me1-coupled 5mC and 5hmC signal more robustly resolves bona fide enhancer states than its unenriched equivalent. These results show that 6B-C&T can be used to evaluate co-occurring epigenetic features and retrieve improved information on individual molecular states (Fig. 1b).”

Discussion (page 5, Discussion paragraph 2):

*“...although all enhancer functional states are marked by H3K4me1, the co-occurrence of 5mC and 5hmC on the same DNA fragment with this histone modification had never been measured. Here, we show that both DNA marks co-occur at significantly higher levels with H3K4me1 than with other histone marks. Our machine learning analysis shows how these different layers of epigenetic information (*i.e.*, *histone and DNA modifications*) are coupled and intrinsically linked with enhancer functional states. Importantly, the use of H3K4me1-specific methylation signatures enables the identification of functional enhancer states without the need for whole-genome sequencing or multiple histone mark enrichment profiles.”*

5. Most of the data is not in the main text, I think it would be useful to have a table(s) to summarise the findings as there is a tremendous amount of data that is skimmed over in the text.

Supplementary Table 8 now contains a summary of all the findings and conclusions in Extended Data.

6. In plots Fig 2b, Ext data Fig 4, and Ext data Fig 6b, the violin plots show a very interesting trend. It's almost as if the %5mC/5hmC is quantised. There are specific bumps of data at regular intervals. The authors should explain why this is, as I'm assuming this cannot be biological so will be a factor of how their experiments were conducted or data processed? What effect does this have on the conclusions?

The nature of 6-base-seq data is discrete since the fraction of (hydroxy)methylation at CpG sites is a function of the (hydroxy)methylation counts and the coverage at each site, both of which are discrete variables (fraction = counts/coverage).

At fixed coverage:

- 1 methylation count covered 20 times = 5%
- 2 counts covered 20 times = 10%

At fixed counts:

- 1 methylation count at a site covered 20 times = 5%
- 1 methylation count at a site covered 40 times = 2.5%

Given the coverage threshold routinely employed for methylation calling (e.g., 10x, 20x) restricts the coverage parameter (denominator) to a very narrow range, the data values (%5hmC or %5mC) become a more direct function of discrete methylation counts (numerator). This produces a “banding” pattern in graphs showing values at individual CpG sites such as scatter plots (see example below) and violin plots. This pattern is absent in plots generated after data smoothing (e.g., window averaging) or descriptive statistics representations (e.g., boxplots).

Extracted from Extended Data Fig. 2c – Illustration of methylation “bands” in 6-base-seq data.

In violin plots generated with *ggplot2* in R, the violin shape is constructed by applying a kernel density estimate (KDE) to the data distribution. This smoothing process can introduce artifacts – such as spurious peaks or apparent gaps – around certain data

Response to reviewers

values, particularly in discrete datasets. As a result, the density envelope may visually exaggerate or underrepresent the presence of data in specific intervals, potentially giving the impression that some intervals contain few or no observations when data points are in fact present. For this reason, we now use stacked bar plots to describe the data in Main Fig. 3b and moved the violin plots to Extended Data Fig. 6 to provide a means of alternative visualisation. Importantly, the above technical observations and visualisation changes do not affect the conclusions in the manuscript.

Response to reviewers.

Reviewer #1 (Remarks to the Author):

Authors addressed concerns raised in the initial submission of the manuscript, with following remaining points:

1. In imprinted gene loci authors demonstrated that e.g. 6B-C&T for H3K4me3 enriches for low 5mC loci, despite close to 50% 5mC present in WG dataset. These are predicted to be the most dramatic examples in the genome, where half of the alleles exist in two completely distinct chromatin and DNA methylation states. While authors decided not to put this figure in the manuscript, I suggest that they should highlight imprinted genes in volcano plots (for at least H3K4me3).

We appreciate the suggestion and have annotated CpG sites from the imprinted gene loci *Igf2r*, *Kcnq1ot* and *Peg3* in the volcano plots in Supplementary Figure 10a and corresponding text:

Supplementary Figure 10a. Differentially methylated and hydroxymethylated CpGs between 6B-C&T and whole-genome 6-base-seq. (a) Volcano plots of differentially methylated 5mCpGs (DMCs, top) and 5hmCpGs (DHMCs, bottom) for each 6B-C&T dataset relative to whole-genome 6-base-seq (WG). Differentially methylated sites ($p\text{-value} < 0.05$, $|\log_2FC| \geq 0.5$) are coloured for each histone mark. Each plot compares histone mark-specific values (6B-C&T) relative to WG 6-base-seq data. Differential methylation was computed from biologically independent experiments (red: H3K27me3, $N = 2$; orange: H3K4me3, $N = 3$; blue: H3K27ac, $N = 3$; green: H3K4me1, $N = 3$; WG, $N = 2$). Non-significant changes are shown in grey. The percentage of significant differences where 6B-C&T values are lower (left) or higher (right) than in whole-genome data are indicated in each plot. CpG sites at example promoter regions showing allele-specific expression in E14TG2A mESCs (imprinted loci) are annotated in each plot (*Igf2r* = white filled circles; *Kcnq1ot* = white filled squares; *Peg3* = white filled triangles).

Text:

“Overall, DMCs predominantly showed lower 5mC levels in 6B-C&T relative to whole-genome (untargeted) data (Supplementary Figure 10a, Supplementary Figure 11). This suggests that DNA associated with histone modifications at these sites is mostly hypomethylated relative to the total pool of DNA fragments. This was

particularly striking in DMCs co-occurring with H3K4me3 at imprinted gene loci³⁹ such as Igfr2, Kcnq1ot1 and Peg3, suggesting that this active histone mark largely associates with the unmethylated allele (Supplementary Figure 10a)."

2. Additional introduction is welcome, but it has a factual error. Statement "5mC is the most abundant cytosine modification (~60% of all Cs in ESCs)⁹ and 5hmC is the next most abundant cytosine modification (less than 5% of all Cs in ESCs)⁹ ..." is incorrect. 5mC constitutes close to 2 - 4% of all Cs. Authors likely meant that 60% of Cs in CpG dinucleotides are methylated, which is closer the actual methylation value. The paper that was referenced does measure modification abundance in CpGs since they perform TLC on labelled nucleotides generated from DNA digestion with MspI, HpaII or Taqα1 – all of them are employed for labelling of Cs in CpGs. Tahiliani et al incorrectly labelled axis in some of their figures, which translated to confusion here.

We correct this oversight (see below). To avoid confusion, we now cite the study by Ito et al. (PMID: 21778364), where the authors quantified the genomic content of all these cytosine modifications using mass spectrometry.

"5mC is the most abundant cytosine modification (2-4% of all Cs in ESCs) and 5hmC is the next most abundant cytosine modification (0.1-0.2% of all Cs in ESCs), with 5fC and 5caC being present at substantially lower levels (~0.002% and ~0.0003%, respectively)⁸."

3. Citations - while listing methods they should also list TAPS (PMID: 30804537) and CAPS (PMID: 33504799) as they are conceptually different from methods that rely on deamination of unmodified Cs (like bisulfite or EM-seq).

We have added the above pyridine borane sequencing methods to the Introduction:

"While early methods such as bisulfite sequencing¹⁹ cannot distinguish 5hmC from 5mC, later approaches – including oxidative bisulfite sequencing²⁰, TET-assisted bisulfite sequencing²¹, APOBEC-coupled epigenetic sequencing²², pyridine borane sequencing methods^{23, 24} and third-generation sequencing platforms (e.g., PacBio, Oxford Nanopore)²⁵ – have enabled base-resolution mapping of both 5mC and 5hmC."

Reviewer #2 (Remarks to the Author):

General Comments [Co-reviewers #2/3]

The authors have thoroughly addressed our original critiques by an improved introduction (with additional background and technology context), addition of essential IgG control reactions, and additional analyses / figures (including reduced claims of enhancer biology in areas of low PTM signal). The rebuttal is a thorough, good-faith effort to address concerns and describe the manuscript revisions. As such we support publication.

Reviewer #3 (Remarks to the Author):

Reviewer #4 (Remarks to the Author):

I am happy my comments have been addressed and supportive of publication